# Small molecule induced oligomerization, clustering and clathrin-independent endocytosis of the dopamine transporter

Tatiana Sorkina, Shiqi Ma, Mads Breum Larsen, Simon C Watkins, Alexander Sorkin*

Department of Cell Biology, University of Pittsburgh School of Medicine, Pittsburgh, United States

**Abstract** Clathrin-independent endocytosis (CIE) mediates internalization of many transmembrane proteins but the mechanisms of cargo recruitment during CIE are poorly understood. We found that the cell-permeable furopyrimidine AIM-100 promotes dramatic oligomerization, clustering and CIE of human and mouse dopamine transporters (DAT), but not of their close homologues, norepinephrine and serotonin transporters. All effects of AIM-100 on DAT and the occupancy of substrate binding sites in the transporter were mutually exclusive, suggesting that AIM-100 may act by binding to DAT. Surprisingly, AIM-100-induced DAT endocytosis was independent of dynamin, cholesterol-rich microdomains and actin cytoskeleton, implying that a novel endocytic mechanism is involved. AIM-100 stimulated trafficking of internalized DAT was also unusual: DAT accumulated in early endosomes without significant recycling or degradation. We propose that AIM-100 augments DAT oligomerization through an allosteric mechanism associated with the DAT conformational state, and that oligomerization-triggered clustering leads to a coat-independent endocytosis and subsequent endosomal retention of DAT.

DOI: https://doi.org/10.7554/eLife.32293.001

*For correspondence:
sorkin@pitt.edu

Competing interests: The authors declare that no competing interests exist.

## Introduction

The activity of plasma membrane receptors, transporters and channels is regulated by endocytosis and post-endocytic trafficking through the endolysosomal system. The general repertoire of endocytic pathways and endosomal sorting events is well described for a number of transmembrane (TM) proteins (endocytic cargo). The molecular mechanisms of cargo internalization via clathrin-mediated endocytosis (CME) are particularly well understood (*Kirchhausen et al., 2014*). The mechanisms underlying the multiple pathways of clathrin-independent endocytosis (CIE) are less defined (*Mayor et al., 2014*). Although progress has been made in characterizing the machinery that allows membrane invagination and vesicle scission during CIE, the mechanisms mediating selective recruitment of the cargo remain elusive. Whether and how cargo itself controls the CIE process is also unclear.

The dopamine transporter (DAT) has served as the prototypic model to study endocytic trafficking of monoamine transporters because of the fundamental physiological and pathophysiological importance of DAT [reviewed in (*Kristensen et al., 2011*; *Melikian, 2004*; *Zahniser and Sorkin, 2009*)]. DAT is responsible for the clearance of dopamine released from synapses, and therefore, controls the duration and amplitude of dopamine signaling in the brain (*Giros et al., 1991*; *Jaber et al., 1997*; *Kristensen et al., 2011*). DAT is also known to be the principal target of psychostimulants like cocaine and amphetamines (*Gainetdinov and Caron, 2003*; *Gowrishankar et al., 2014*; *Spiga et al., 2008*; *Volkow and Morales, 2015*; *Willuhn et al., 2010*; *Wise, 2008*). DAT belongs to the high-affinity saline carrier (SLC) six gene family of Na$^+$, Cl$^-$-dependent transporters

consisting of 12 TM domains, and cytosolic N- and C-terminal tails (*Kristensen et al., 2011*). DAT is proposed to form dimers and high-order oligomers, although the mechanisms of oligomerization are not understood (*Hastrup et al., 2001*; *Hastrup et al., 2003*; *Li et al., 2010*; *Sorkina et al., 2003*; *Torres et al., 2003*). It has been proposed that DAT dimerization/multimerization modulates the substrate transport activity of DAT and is necessary for the effective transport of newly-synthesized DAT from the endoplasmic reticulum (*Chen and Reith, 2008*; *Sorkina et al., 2003*; *Torres et al., 2003*; *Zhen et al., 2015*; *Zhen and Reith, 2018*).

Regulation of DAT function by endocytic trafficking has been demonstrated in heterologous expression systems and in dopaminergic neurons. DAT has been shown to internalize in response to protein kinase C (PKC) activation, amphetamines, substrates, glial cell line-derived neurotrophic factor, neuronal activity and inhibition of Ack1 (Activated Cdc42 Kinase) (*Eriksen et al., 2009*; *Gabriel et al., 2013*; *Hoover et al., 2007*; *Huff et al., 1997*; *Melikian and Buckley, 1999*; *Rao et al., 2012*; *Richardson et al., 2016*; *Sorkina et al., 2003*; *Vaughan et al., 1997*; *Wheeler et al., 2015*; *Wu et al., 2015*; *Zhu et al., 2015*). DAT appears to be capable of internalization through both CME and CIE (*Cremona et al., 2011*; *Gabriel et al., 2013*; *Sorkina et al., 2013*; *Sorkina et al., 2005*; *Wheeler et al., 2015*; *Wu et al., 2015*). Although transporters of the SLC6 family share the molecular fold and conformational transition mechanics during substrate transport, some of the trafficking mechanisms discovered for DAT have yet to be demonstrated for other SLC6 transporters (*Kristensen et al., 2011*; *Matthies et al., 2010*; *Vuorenpää et al., 2016b*; *Wu et al., 2015*), suggesting there may be cargo-specific mechanisms controlling endocytic trafficking of these transporters.

It is possible that DAT endocytosis induced by the Ack1 inhibitor AIM-100 (5,6-Diphenyl-N-[[(2S)-tetrahydro-2-furanyl]methyl]furo[2,3-d]pyrimidin-4-amine), could be an example of a cargo (DAT)-specific endocytosis pathway because the same compound does not increase endocytosis of the close homolog of DAT, serotonin transporter (SERT) (*Wu et al., 2015*). Hence, we compared the mechanisms of AIM-100-induced endocytosis of DAT with those of its constitutive and PKC-dependent endocytosis. Unexpectedly, multiple independent approaches demonstrated clearly that AIM-100 induces robust oligomerization and clustering of DAT, which precedes clathrin-independent internalization of DAT, and that these effects of AIM-100 are specific to DAT but unrelated to Ack1. Altogether, the present findings led to a hypothetical model whereby AIM-100 acts directly on DAT via an allosteric mechanism, and uncovered a novel CIE mechanism driven by the conformation-coupled cargo oligomerization.

## Results

### AIM-100 induces DAT oligomerization and endocytosis in heterologous cells and dopaminergic neurons

Recent studies by Wu and co-workers used biotinylation techniques to demonstrate that AIM-100, an Ack1 inhibitor, promotes internalization of DAT in SK-N-MC cells and mouse brain striatal slices (*Wu et al., 2015*). To gain insight into the mechanism of this novel endocytic regulation of DAT, we analyzed AIM-100-induced trafficking of DAT fused to yellow fluorescent protein (YFP) (YFP-DAT) (*Sorkina et al., 2003*) and YFP-DAT additionally tagged with an extracellular hemagglutinin (HA) epitope (YFP-HA-DAT) (*Sorkina et al., 2006*) using quantitative fluorescence microscopy. Two approaches were employed and produced essentially identical results using YFP-DAT, YFP-HA-DAT or endogenous HA-DAT: (i) direct YFP imaging in fixed and living cells; and (ii) immunofluorescence imaging using an HA antibody uptake assay (*Sorkina et al., 2006*). Direct YFP imaging demonstrated that treatment of porcine aortic endothelial (PAE) cells stably expressing YFP-HA-DAT with AIM-100 at 37°C resulted in dramatic re-distribution of YFP-HA-DAT from the plasma membrane, where the transporter is concentrated in filopodia, cell edges and ruffles, to intracellular vesicular compartments (*Figure 1A*). AIM-100-induced vesicular accumulation of YFP-HA-DAT was observed at AIM-100 concentrations $\geq$ 10 μM and reached a maximal extent at 20 μM (*Figure 1B*). Quantitative analysis of the time-course of YFP-HA-DAT endocytosis induced by AIM-100 using an HA antibody uptake assay revealed that this endocytosis follows a linear kinetics without reaching a plateau for at least 2 hr (*Figure 1C–D*). Such kinetics is characteristic of a non-saturable and/or irreversible trafficking process.

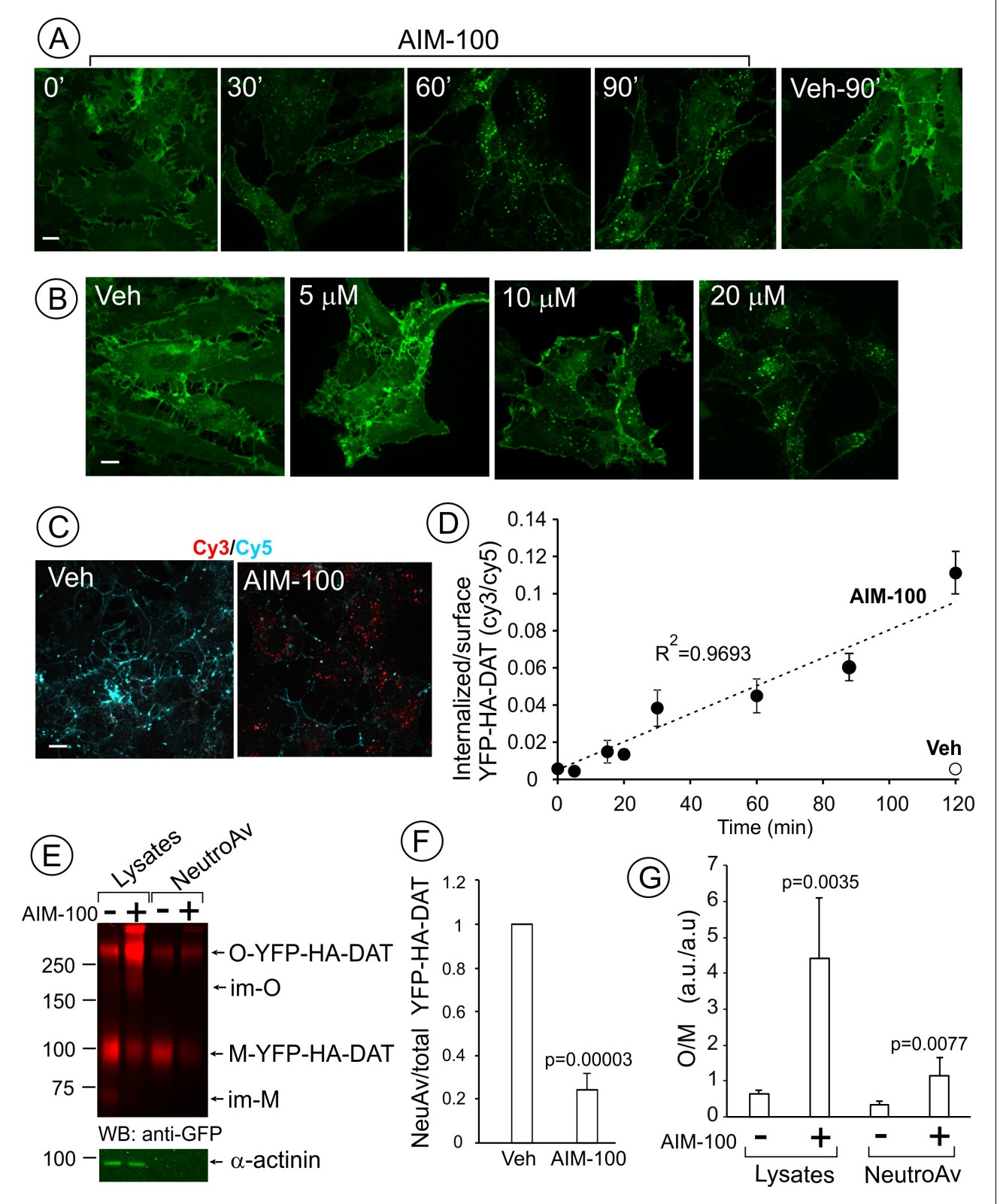

**Figure 1.** AIM-100 causes accumulation of YFP-HA-DAT in endosomes and enhances DAT oligomerization in PAE cells. (**A**) Cells were incubated with vehicle (DMSO) or 20 μM AIM-100 for 0–90 min at 37°C, and 3D images were acquired from fixed cells through the 515 nm channel. Maximal intensity projections of all z-planes of representative images are shown. Scale bar, 10 μm. (**B**) Cells were incubated with vehicle (*veh*, DMSO) or 5–20 μM AIM-100 for 90 min at 37°C and fixed. Maximal intensity projections of all z-planes of representative YFP images are shown. Scale bar, 10 μm. (**C**) Cells

*Figure 1 continued on next page*

*Figure 1 continued*

were pre-incubated with HA11 for 1 hr at 37°C, and then further incubated with vehicle or 20 µM AIM-100 for 0–120 min at 37°C. After fixation, cultures were stained with secondary anti-mouse antibodies conjugated with Cy5 (*surface HA-DAT*), permeabilized with Triton X-100 and stained with secondary anti-mouse conjugated with Cy3 (*internalized HA-DAT*). 3D images were acquired through 488 (YFP, not shown), 561 (Cy3, *red*) and 640 nm (Cy5, *cyan*) channels. Individual confocal sections of 120 min time points are shown. (D) Time course of the Cy3/Cy5 ratio was calculated in HA11 uptake experiments exemplified in (C). Results are presented as mean values (±S.D., n = 3). $R^2$, linearity coefficient. (E) Cells were incubated with vehicle or 20 µM AIM-100 for 2 hr at 37°C, and biotinylated at 4°C as described in 'Methods'. An aliquot of lysates was incubated with NeutroAvidin (NeuAv) to pull-down biotinylated proteins. Lysates and pulldowns were resolved by SDS-PAGE electrophoresis and probed by western blotting with GFP and α-actinin (loading control) antibodies. O, oligomers; M, monomers; im-M, immature monomers; im-O, immature oligomers. (F) Quantification of the fraction of the biotinylated DAT relative to total DAT (lysates) in six independent experiments (mean ±S.D.). P values are 'AIM-100' compared to 'vehicle'. (G) Quantification of the ratio of the amount of >250 kDa species (O, oligomers,) to the amount of 95 kDa YFP-HA-DAT species (M, monomers). Results are presented as mean values (±SD, n = 6). P values are for 'AIM-100' compared to 'vehicle'.

DOI: https://doi.org/10.7554/eLife.32293.002

The following figure supplements are available for figure 1:

**Figure supplement 1.** Biochemical properties of SDS-resistant DAT oligomers.

DOI: https://doi.org/10.7554/eLife.32293.003

**Figure supplement 2.** AIM-100-induced DAT endocytosis in HeLa, HEK293 and SH-SY5Y cells.

DOI: https://doi.org/10.7554/eLife.32293.004

Cell-surface biotinylation experiments confirmed down-regulation of the plasma membrane YPF-HA-DAT by approximately 75% after a 2 hr AIM-100 treatment of PAE cells (*Figure 1E–F*). Surprisingly, a greater than 5-fold increase in the fraction of high molecular mass species (~290 kDa) of YFP-HA-DAT was observed in lysates of AIM-100-treated cells when compared with vehicle-treated cells (*Figure 1E and G*). The same, discrete band of a high molecular mass DAT species could also be detected consistently in the absence of AIM-100, although the relative fraction of this species (10–40%) varies between different experiments and between cell lines. AIM-100 substantially increased the amount of this species, and in addition, the abundance of the larger DAT species in all experiments, suggesting that AIM-100 stabilizes SDS-resistant DAT oligomers and higher-order oligomers. Because AIM-100-induced oligomers were not disrupted by β-mercaptoethanol, iodoacetamide (see 'Methods') and dithiothreitol (DTT) (*Figure 1—figure supplement 1*), oligomerization was not due to formation of disulfide bonds between DAT molecules. The ratio of oligomers to monomers (O/M) in the biotinylated fraction was significantly increased by AIM-100 demonstrating that AIM-100 induces DAT oligomerization at the cell surface (*Figure 1G*). However, the O/M ratio was 4-times lower in the biotinylated fraction than in total cell lysates (surface plus endosomal DAT) in AIM-100-treated cells. This data indicate that AIM-100-induced DAT oligomers are enriched in endosomes as compared to the plasma membrane, and suggest that oligomerization facilitates DAT endocytosis. Small amounts of immature monomeric (~70 kDa) and corresponding oligomeric YFP-HA-DAT (~200 kDa) were detected in lysates of untreated and AIM-100-treated cells, respectively, but not in the biotinylated fractions (*Figure 1E*), indicating that AIM-100 also causes oligomerization of newly-synthesized DAT located in the endoplasmic reticulum.

AIM-100 also induced endocytosis of YFP-HA-DAT or untagged DAT stably expressed in HeLa, HEK293 and human neuroblastoma SH-SY5Y cells (*Figure 1—figure supplement 2*). Moreover, primary cultures of post-natal mesencephalic dopaminergic neurons derived from HA-DAT knock-in mouse displayed strong endocytosis of HA-DAT following AIM-100 treatment, as revealed using the HA11 antibody uptake assay (*Figure 2*). Cy3-enriched vesicles (internalized HA11:HA-DAT complexes) were abundant in axonal processes, varicosities, axonal widenings and soma of AIM-100-exposed neurons decorated with Cy5 (cell-surface HA11:HA-DAT complexes) (*Figure 2A–B*). The Cy3/Cy5 ratio (the extent of endocytosis) was several times higher in AIM-100-treated than in control neurons (*Figure 2C*). Finally, AIM-100 treatment of striatal synaptosomes prepared from adult HA-DAT mice substantially increased HA-DAT oligomerization (*Figure 2D*). Essentially, *Figure 1*, *Figure 1—figure supplement 2*, and *Figure 2* all demonstrate similar effects of AIM-100 on the endocytic traffic and oligomerization of endogenous and heterologously expressed DAT. Therefore, in subsequent experiments presented below in *Figures 3–11* we used our collection of PAE cells stably expressing wild-type and mutant DATs. These cells are amenable to an efficient DNA and siRNA transfection, and a biochemical analysis (*Sorkina et al., 2003*; *Sorkina et al., 2005*; *Sorkina et al.,*

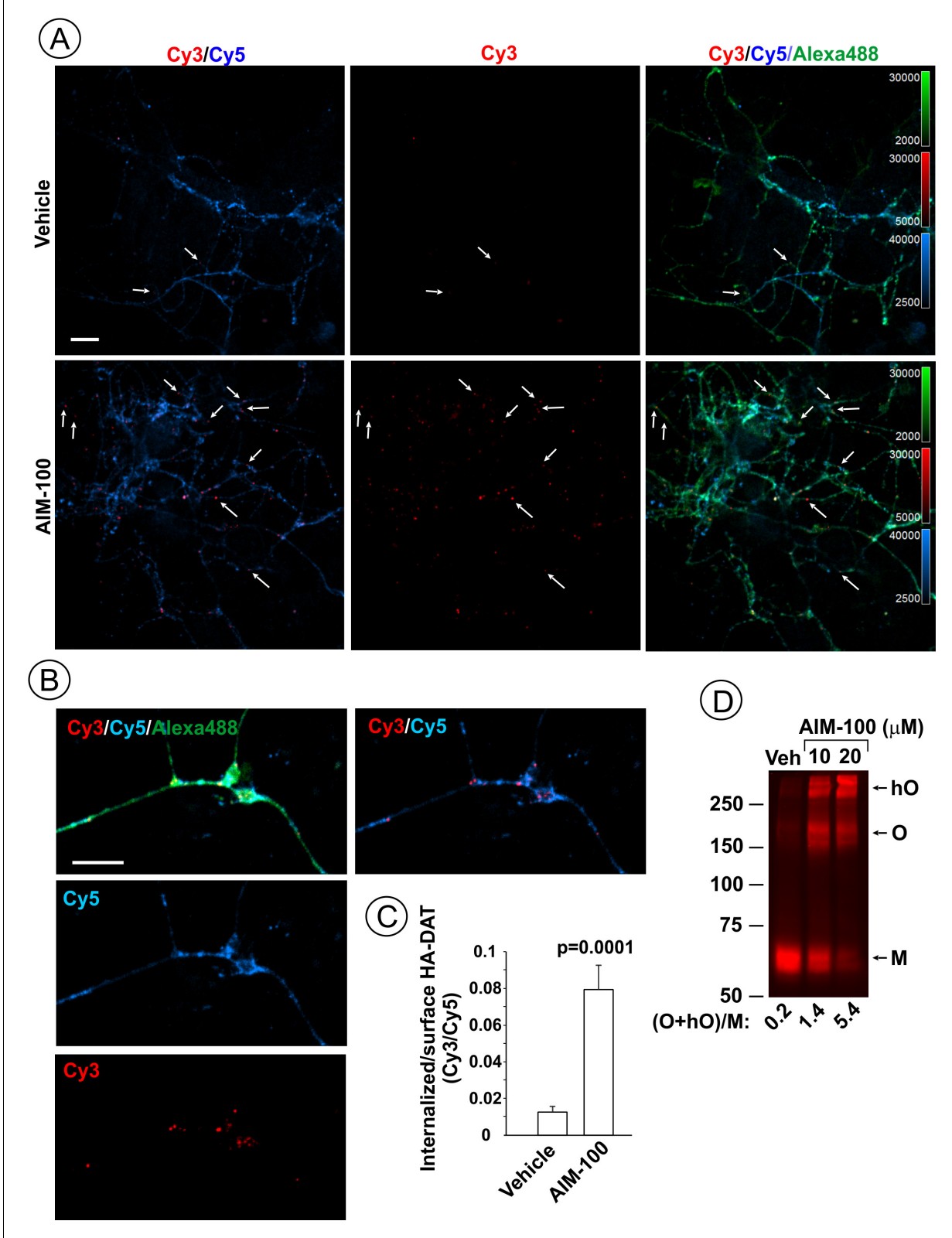

**Figure 2.** AIM-100 induces endogenous HA-DAT endocytosis and oligomerization in mouse dopaminergic neurons. (**A–C**) Cultured post-natal mesencephalic neuronal cultures were pre-incubated with HA11 antibodies for 30 min at 37°C, and then further incubated with vehicle (DMSO) or 10 μM AIM-100 for 2 hr at 37°C. After fixation, cultures were stained with secondary anti-mouse antibodies conjugated with Cy5 (*surface HA-DAT*), permeabilized with Triton X-100 and incubated with rat-anti-DAT antibody, and then stained with secondary anti-mouse conjugated with Cy3

*Figure 2 continued on next page*

*Figure 2 continued*

(*internalized HA-DAT*) and secondary anti-rat antibody conjugated with Alexa488 (total HA-DAT immunoreactivity). 3D images were acquired through 488 (Alexa488, *green*), 561 (Cy3, *red*) and 640 nm (Cy5, *blue*) channels. Individual confocal sections are presented. (**A**) Representative images. Arrows point on examples of Cy3 enriched puncta (endosomes) that localize in neuronal processes. (**B**) Representative images of endosomal HA-DAT (Cy3 fluorescence) in axonal varicosities in AIM-100-treated cells. Scale bars, 10 µm. (**C**) Quantification of Cy3/Cy5 ratio values from 3D images exemplified in (**A**). Results are presented as mean values (±S.E.M.; n = 13-15). P value is calculated for 'AIM-100' compared to 'vehicle'. (**D**) Striatal synaptosomes prepared from adult HA-DAT mice were incubated with vehicle or 10–20 µM AIM-100 for 2 hr at 37°C. Lysates were resolved by electrophoresis and probed by western blotting with DAT antibody. *M*, monomers, *O*, oligomers; *hO*, high-order oligomers. O/M ratios are presented as a mean of the ratio values obtained from two mice.

DOI: https://doi.org/10.7554/eLife.32293.005

*2006*; *Sorkina et al., 2009*). Experimental conditions of the maximally strong effects of AIM-100 (20 µM; 60–120 min at 37°C) (*Figure 1A and B*) were utilized to study the mechanisms of AIM-100 effects on DAT.

## AIM-100 induced rapid nanoclustering of DAT on the cell surface

To analyze the process of AIM-100-induced DAT oligomerization in living cells, total internal reflection fluorescence microscopy (TIR-FM) was employed. Treatment of PAE/YFP-HA-DAT cells with AIM-100 resulted in a dramatic increase in the number of diffraction-limited (<200 nm) spots (clusters) of YFP-HA-DAT on the basal cell surface (*Figure 3A–B*, and *Figure 3—video 1*). The number of spots approached a maximum after ~10–20 min of AIM-100 treatment and gradually declined during a subsequent 30 min incubation (*Figure 3B*). Approximately 35% of spots persisted for >1 hr, whereas other spots were quite transient with a life-time of a few minutes, putative of endocytic uptake (*Figure 3C*). We noticed some spots were moving laterally within the imaging plane, characteristic of newly-formed endocytic vesicles. In addition to an almost complete clustering of YFP-HA-DAT in the planar regions of the membrane, the transporter either moved away from filopodia or formed clusters within the filopodia (*Figure 3D*). Simultaneously, strong accumulation of YFP-HA-DAT at the cell edges was observed (*Figure 3A*; *Figure 3—video 1*). Importantly, the pool of YFP-HA-DAT visualized by TIR-FM is relatively small compared to the total cellular YFP-HA-DAT owing to high sensitivity of TIR-FM. Three-dimensional confocal imaging demonstrated that although small clusters of YFP-HA-DAT were seen on both the bottom and top plasma membranes domains of the cell, a bulk of total cellular YFP-HA-DAT was accumulated in endosomes after a 45 min AIM-100 treatment (*Figure 3E*). Given the observed timing of cluster formation on the cell surface (*Figure 3B*), and the kinetics of DAT accumulation in endosomes (*Figure 1C*), it would appear that AIM-100-induced clustering of DAT precedes its endocytosis.

To directly examine the effect of AIM-100 on DAT oligomerization in living cells, fluorescence resonance energy transfer (FRET) microscopy was used. Constitutive oligomerization of DAT has been previously demonstrated by FRET (*Sorkina et al., 2003*). Using the same method of sensitized FRET measuremnts in cells co-expressing CFP-DAT and YFP-DAT, we found that the intensity of FRET from CFP to YFP was significantly higher in the plasma membrane clusters and intracellular vesicles in AIM-100-treated cells than this intensity in vehicle-treated cells where FRET was measured in various structures and regions of DAT accumulation, such as filopodia, ruffles and cell-cell contacts (*Figure 3—figure supplement 1*). An increased FRET in AIM-100-treated cells is indicative of a higher probability or an increased stability of the interaction between DAT molecules, and/or a closer proximity of multiple DAT molecules, thus demonstrating AIM-100-induced DAT oligomerization in living cells.

## AIM-100 does not induce oligomerization and endocytosis of NET or SERT

It has been shown that AIM-100 does not increase the endocytosis of SERT in SK-N-MC cells (*Wu et al., 2015*). Therefore, we compared AIM-100 effects on the closest homolog of DAT, norepinephrine transporter (NET) with AIM-100 effects on DAT and SERT. Live-cell imaging of PAE cells either transiently or stably expressing GFP-NET or GFP-SERT revealed that both transporters are efficiently delivered to the plasma membrane. In contrast to its effects on DAT, AIM-100 did not induce clustering and endocytosis of GFP-NET and GFP-SERT (*Figure 4A*). Moreover, GFP-SERT

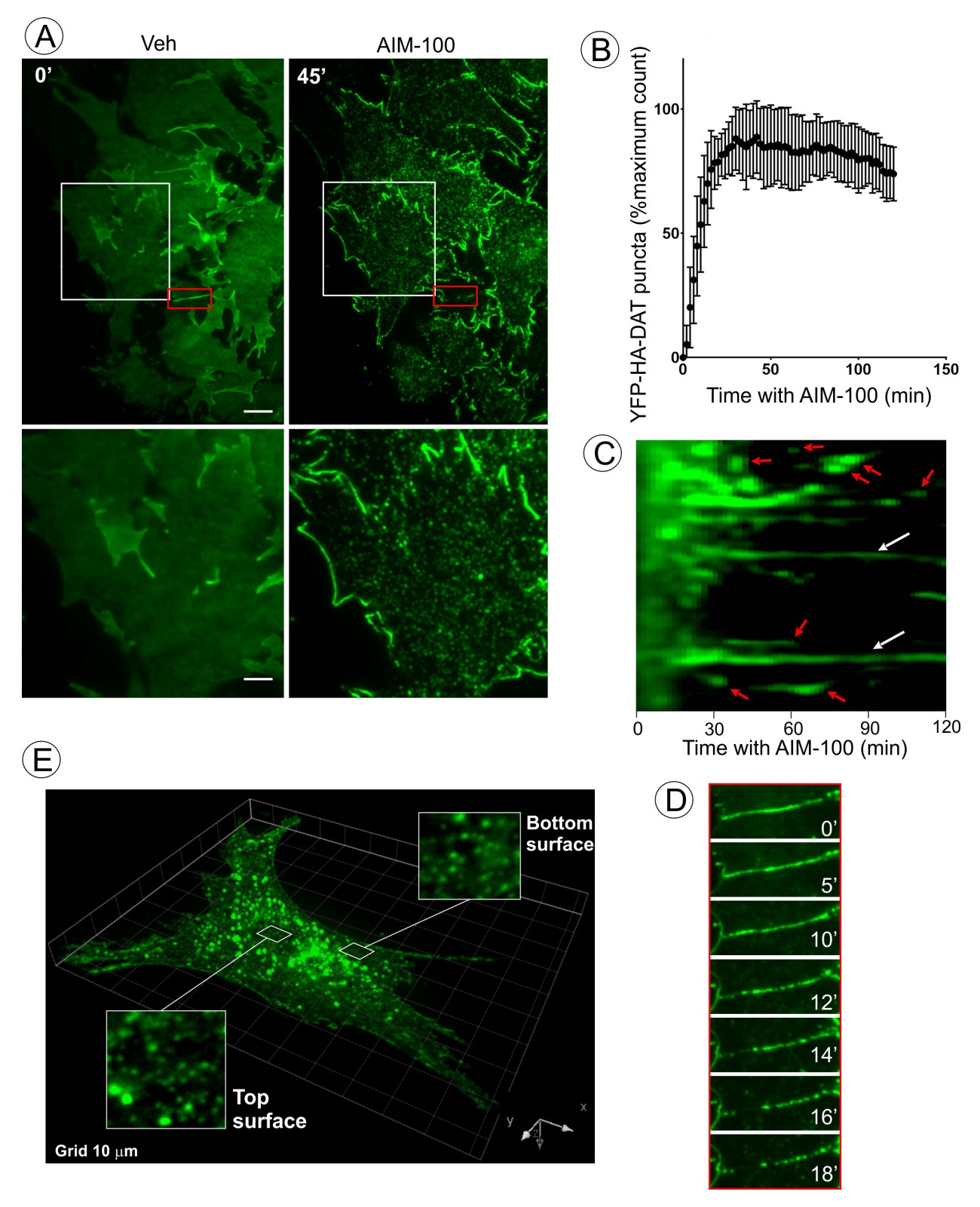

**Figure 3.** AIM-100 causes formation of DAT nanoclusters on the plasma membrane. (**A**) Time-lapse TIR-FM imaging of PAE/YFP-HA-DAT cells. Individual time frames before and 45 min after addition of AIM-100 (20 μM) are shown. (See *Figure 3—video 1*). Insets below represent high magnification images of the regions marked by white rectangles. Scale bars are 5 μm in images and 2 μm in insets. (**B**) The number of YFP-HA-DAT puncta in cells treated with AIM-100 was calculated as described in 'Materials and methods' in nine time-lapse TIR-FM imaging sequences represented

*Figure 3 continued on next page*

*Figure 3 continued*

in (**A**). Mean values (±S.D.) of the percentage of maximum number of puncta per time frame are presented. (**C**) Representative kymographs time-series of YFP-HA-DAT imaging from randomly-selected region of cells presented in (**A**). Examples of stable clusters are indicated by white arrows. Examples of shorter-living clusters that may represent vesicle scission events are shown by red arrows. (**D**) High magnification time-lapse images (0–18 min with AIM-100) of the region marked by red rectangles in (**A**) demonstrating clustering of YFP-HA-DAT in a filopodium. (**E**) Cell were incubated with AIM-100 for 45 min as in (**A**), and 3D stack of confocal images were acquired. Insets show high magnification images of regions of the bottom and top of the cell (above nucleus) indicated by white rectangles.

DOI: https://doi.org/10.7554/eLife.32293.006

The following video and figure supplement are available for figure 3:

**Figure supplement 1.** AIM-100 increases FRET between CFP- and YFP-DATs.

DOI: https://doi.org/10.7554/eLife.32293.007

**Figure 3—video 1.** Time-lapse TIR-FM imaging of PAE/YFP-HA-DAT cells.

DOI: https://doi.org/10.7554/eLife.32293.008

and GFP-NET were detected on western blots as predominantly monomeric species in the absence or presence of AIM-100 (*Figure 4B*). These data demonstrate that the specificity of AIM-100 effects on transporter endocytosis correlates with the specific effects of AIM-100 on transporter oligomerization.

## DAT inhibitors and substrates diminish the effects of AIM-100 on DAT

AIM-100 has been shown to inhibit binding of a radiolabeled cocaine analog to DAT, and given the structural similarity of AIM-100 to cocaine and DAT substrates, AIM-100 was proposed to bind DAT (*Wu et al., 2015*). We therefore tested the effect of cocaine, a competitive inhibitor of DAT, on AIM-100-stimulated DAT oligomerization and endocytosis. Treatment of cells with cocaine (1–10 µM) significantly inhibited AIM-100-enhanced oligomerization, clustering in the plasma membrane and endocytosis of YFP-DAT in cells treated with AIM-100 (*Figure 5A–E*). These data suggest that the effects of AIM-100 on DAT are prevented by the occupancy of S1 and S2 binding sites in the DAT or by the outward-facing (OF) conformation of DAT which is stabilized by cocaine binding (*Loland et al., 2002*; *Reith et al., 2001*). To define the role of substrate binding sites and DAT conformation, the effects of AIM-100 on the W63A mutant of YFP-HA-DAT were examined. W63 is essential for maintaining the network of intramolecular interactions supporting the OF conformation of DAT (*Penmatsa et al., 2013*). Therefore, the W63A mutation results in narrowing the extracellular vestibule, which leads to the loss of substrate transport and cocaine binding, and a higher probability of an inward-facing (IF) conformation (*Chen et al., 2001*; *Ma et al., 2017*; *Sorkina et al., 2009*). AIM-100 triggered the robust oligomerization and endocytosis of the W63A mutant but in this case endocytosis was unaffected by cocaine (*Figure 5F–H*). These observations imply that (i) the AIM-100 effects on DAT do not require accessible extracellular substrate binding sites per se; but that (ii) cocaine binding to these sites in the wild-type DAT is necessary for counteracting the effects of AIM-100 on DAT.

To further examine the importance of DAT conformation in the observed effects of AIM-100, we tested whether DAT substrates interfere with the AIM-100 effects on DAT. Computational modeling predicts that the conformational equilibrium of DAT is shifted towards occluded-intermediate (OC) and IF states in the presence of substrates (*Cheng and Bahar, 2015*). This prediction is supported by the experimental evidence in studies of SERT (*Zhang et al., 2016*). We found that oligomerization and endocytosis of DAT in AIM-100-treated cells were significantly decreased by saturating concentrations (20–100 µM) of DAT substrates, dopamine (DA) and amphetamine (Amph) (*Figure 6*). Moreover, modafinil, an atypical DAT inhibitor, that has been proposed to bind to S1 and S2 sites with micromolar affinity and stabilize DAT in the OC and/or IF states (*Schmitt and Reith, 2011*), also significantly reduced AIM-100-induced DAT oligomerization and endocytosis (*Figure 6*). These experiments provide additional evidence that the occupancy of substrate binding sites in wild-type DAT is inhibitory to the AIM-100 effects on DAT and suggest that the shift to an OC/IF conformation is not sufficient for AIM-100 induced oligomerization and endocytosis of the wild-type DAT when substrates or inhibitors are present.

Together, the data presented in *Figures 5* and *6* imply that if the effects of AIM-100 on DAT are indeed mediated through the direct binding of AIM-100 to DAT that is blocked by DAT inhibitors

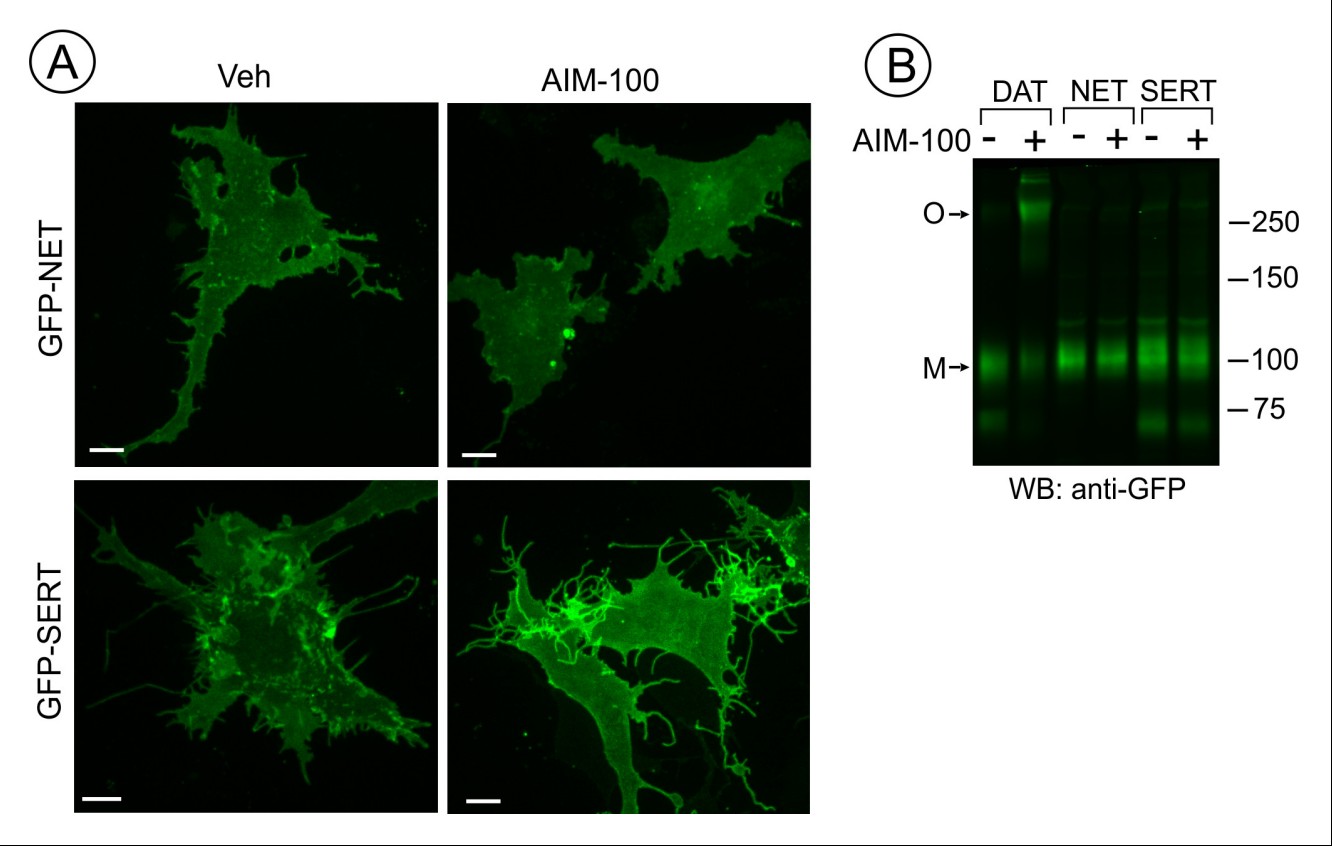

**Figure 4.** SERT and NET are not endocytosed and oligomerized in AIM-100 treated cells. (**A**) PAE cells transiently or stably expressing GFP-SERT or GFP-NET were incubated with vehicle or 20 μM AIM-100 for 2 hr at 37°C. 3D images were acquired from living cells. Maximal intensity projections of z-planes of representative YFP images are shown. Scale bars, 10 μm. (**B**) PAE/cells stably expressing YFP-DAT, GFP-SERT or GFP-NET were incubated as in (**A**), and lysates were electrophoresed and probed by western blotting with the GFP antibody. *M*, monomers; *O*, oligomers.
DOI: https://doi.org/10.7554/eLife.32293.009

and substrates, there should be a reciprocal inhibition of binding of these compounds to DAT by AIM-100. Such inhibition of cocaine binding to DAT by AIM-100 (Ki ~20–40 μM) has been previously demonstrated (*Wu et al., 2015*). To test whether the observed inhibitory effect of AIM-100 is competitive or non-competitive in nature, binding parameters of a cocaine analog, [3-H](2)−2-β-carbomethoxy-3-β-(4-fluorophenyl)tropane ([3 hr]-CFT) were determined in the absence or presence of AIM-100 (*Figure 5—figure supplement 1A–B*). The binding assays were performed under the experimental conditions used to demonstrate AIM-100 effects on DAT oligomerization, with the exception that binding was carried out at 20°C to minimize DAT endocytosis, which was confirmed by live-cell imaging of parallel cells. DAT oligomerization was significantly increased by AIM-100 at 20°C albeit to a lesser extent than at 37°C (*Figure 5—figure supplement 1C*). In five independent experiments AIM-100 (20 μM) consistently and statistically significantly decreased the Bmax value (an average decrease of 26%) but did not significantly affect $K_D$ values (*Figure 5—figure supplement 1A–B*). Thus, the effect of AIM-100 on CFT binding parameters is consistent with the hypothesis that this effect involves an allosteric mechanism. While the precise molecular mechanisms of AIM-100 binding to DAT remain to be defined, the data in *Figures 5–6* further support the strong correlation between DAT oligomerization, clustering and endocytosis stimulated by AIM-100.

## DAT accumulates in early/sorting but not recycling or late endosomes in AIM-100-treated cells

To define DAT localization after AIM-100-induced endocytosis, fluorescently tagged Rab5, EEA.1, Rab11 and Rab7 were transiently expressed in PAE/YFP-HA-DAT cells. In AIM-100-treated cells YFP-

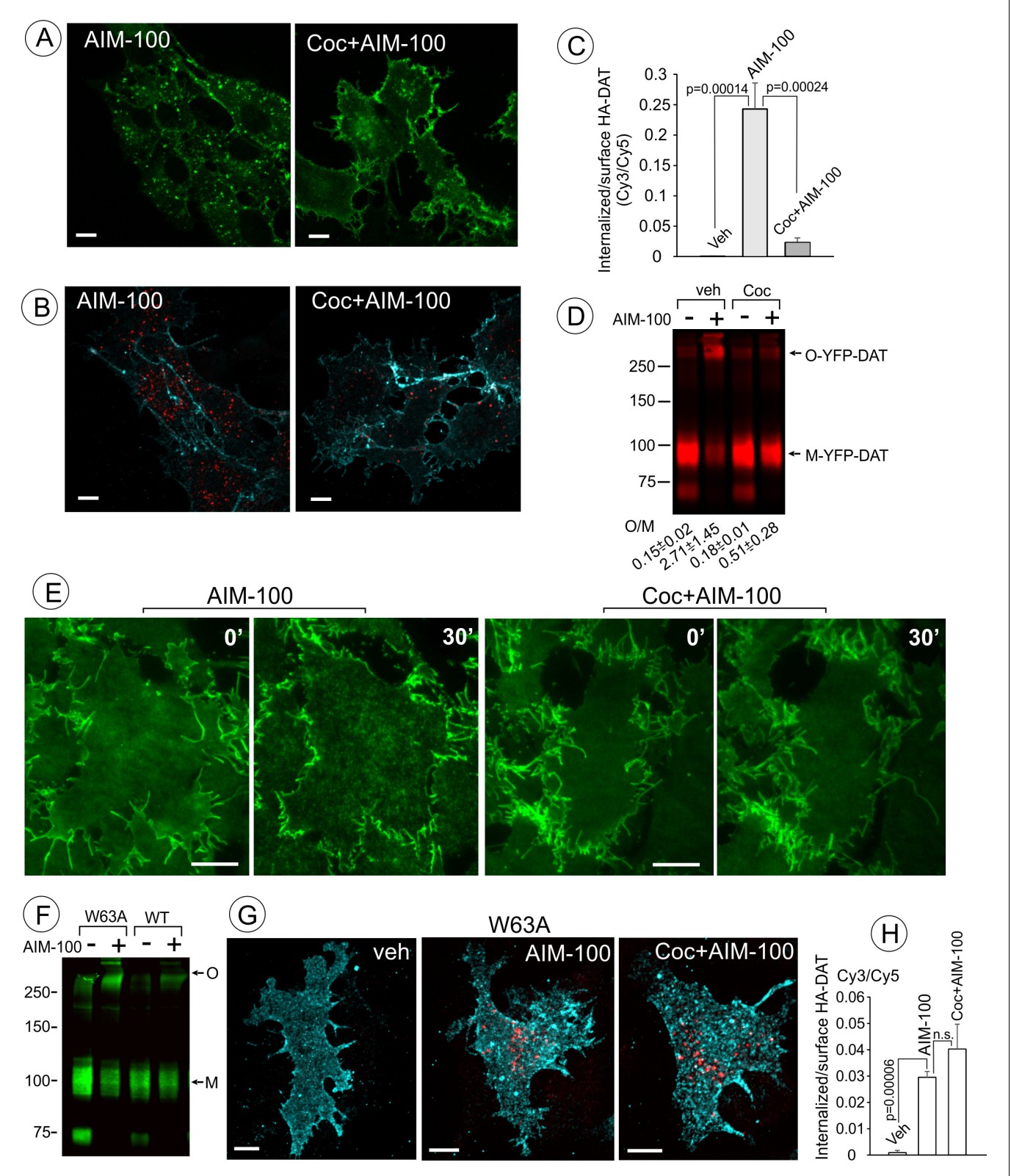

**Figure 5.** Cocaine inhibits AIM-100-induced endocytosis, oligomerization and clustering of DAT. (**A**) PAE/YFP-HA-DAT cells were incubated without or with 4 μM cocaine for 30 min, and further incubated with vehicle (DMSO) or 20 μM AIM-100 for 1.5 hr at 37°C in the same media. 3D images were acquired from living cells through the 515 channel. Individual confocal sections through the middle of the cell are shown. (**B**) PAE/YFP-HA-DAT cells were incubated with HA11 for 30 min at 37°C, and then incubated with vehicle (PBS) or 10 μM cocaine (*coc*) for 10 min, and further incubated with

*Figure 5 continued on next page*

*Figure 5 continued*

vehicle (DMSO) or 20 μM AIM-100 in the presence of vehicle (PBS) or 1 μM cocaine for 2 hr at 37°C. After fixation, cultures were stained with secondary anti-mouse antibodies conjugated with Cy5 (*surface HA-DAT*), permeabilized with Triton X-100 and stained with secondary anti-mouse conjugated with Cy3 (*internalized HA-DAT*). 3D images were acquired through 488 (YFP, not shown), 561 (Cy3, *red*) and 640 nm (Cy5, *cyan*) channels. Individual confocal sections are presented. (**C**) Cy3/Cy5 ratios were calculated in experiments exemplified in (**B**). Results are presented as mean values of the ratio (±S.D.; n = 3-5). (**D**) PAE/YFP-DAT cells were incubated with vehicle or 10 μM cocaine for 10 min at 37°C, and further incubated with vehicle or 20 μM AIM-100 for 2 hr at 37°C, and lysates were analyzed by immunoblotting with the GFP antibody. Representative experiment is shown. *M*, monomers; *O*, oligomers. The mean values of the O/M ratios (±S.D.) (below the blot) were measured in three independent experiments. (**E**) Time-lapse TIR-FM imaging of PAE/YFP-HA-DAT cells that were pre-incubated or not with 10 μM cocaine for 5 min at 37°C, and then incubated with AIM-100 (20 μM) for 1 hr at 37°C. Individual time frames before (0') and 30 min after addition of AIM-100 are shown. Scale bars are 10 μm. (**F**) PAE/YFP-DAT and PAE/W63A-YFP-HA-DAT cells were incubated with DMSO (veh) or AIM-100 as in (**D**), and lysates were analyzed by western blotting with the GFP antibody. Representative experiment of three independent experiments is shown. *M*, monomers; *O*, oligomers. (**G**) PAE/W63A-YFP-HA-DAT cells were incubated with HA11, cocaine and AIM-100 as in (**B**). After fixation, cultures were stained with secondary antibodies as in (**B**). 3D images were acquired through 488 (YFP, not shown), 561 (Cy3, red) and 640 nm (Cy5, cyan) channels. Maximum intensity projections of z-stack of confocal images are presented. Scale bars, 10 μm. (**H**) Cy3/Cy5 ratios were calculated in experiments exemplified in (**E**). Results are presented as mean values of the ratio (±SD, n = 4-5). n.s., p=0.146.

DOI: https://doi.org/10.7554/eLife.32293.010

The following figure supplement is available for figure 5:

**Figure supplement 1.** AIM-100 non-competitively inhibits CFT binding.

DOI: https://doi.org/10.7554/eLife.32293.011

HA-DAT was highly co-localized with Rab5 and EEA.1, markers of early and sorting endosomes, but not with Rab11 (*Figure 7A*), suggesting that DAT is not efficiently sorted to the recycling compartment. Strong co-localization of DAT with endogenous EEA.1 was also confirmed using immunofluorescence staining of fixed cells (*Figure 7B*). Only a minimal amount of YFP-HA-DAT was detected in Rab7- and LysoTrackerRed-labeled compartments, indicating that DAT is not sorted to late endosomes and lysosomes after AIM-100-induced internalization (*Figure 8A*). AIM-100 did not induce degradation of YFP-HA-DAT or YFP-DAT in contrast to a substantial PKC-dependent degradation of the transporter in cells treated with phorbol 12-myristate 13-acetate (PMA) (*Figure 8B–C*). This result is consistent with the observation that AIM-100 does not significantly increase YFP-HA-DAT ubiquitination (*Figure 8—figure supplement 1*). The data in *Figures 7–8* reveal an unconventional itinerary of AIM-100-induced DAT traffic that involves the continuous retention of internalized DAT in early/sorting endosomes.

## DAT internalized in AIM-100-treated cells does not recycle

The linear kinetics of AIM-100-induced DAT accumulation in early endosomes suggests that DAT may not recycle back to the plasma membrane from these endosomes. To test this hypothesis, YFP-HA-DAT was labeled with HA11 and internalized in the presence of AIM-100 for 1.5 hr followed by washing and further incubating the cells at 37°C without AIM-100. This 1.5 hr chase incubation did not lead to re-distribution of YFP-HA-DAT from endosomes to the cell surface (recycling), but unexpectedly, actually resulted in an augmented accumulation of YFP-HA-DAT in endosomes (*Figure 9*). Cocaine did not prevent additional endocytosis of HA11:YFP-HA-DAT complexes during the chase incubation (*Figure 9*), suggesting that cocaine binding must precede AIM-100 treatment in order for cocaine to inhibit AIM-100-induced DAT endocytosis. Importantly, these data indicate that AIM-100 has effects on DAT that are not readily reversible, and that DAT internalized in the presence of AIM-100 is not measurably recycled. In contrast, AIM-100 did not inhibit recycling of DAT that had been internalized upon PKC activation and protected from AIM-100-induced endocytosis by cocaine (*Figure 9—figure supplement 1*). These data suggest that AIM-100 does not affect the recycling process per se but acts in a cargo-specific fashion by blocking recycling of DAT exclusively only if DAT internalization is induced by AIM-100. The data in *Figure 9* together with the observations of (i) linear kinetics of DAT accumulation in endosomes (*Figure 1C*); and (ii) lack of DAT in Rab11 recycling endosomes (*Figure 7*) suggest that the robust AIM-100-induced accumulation of DAT in endosomes is at least partially due to an inability of DAT to recycle in the presence of AIM-100.

The endosomal accumulation of DAT has also been observed when DAT recycling is inhibited by monensin (*Sorkina et al., 2005*). However, the extent of endosomal accumulation of DAT due to its

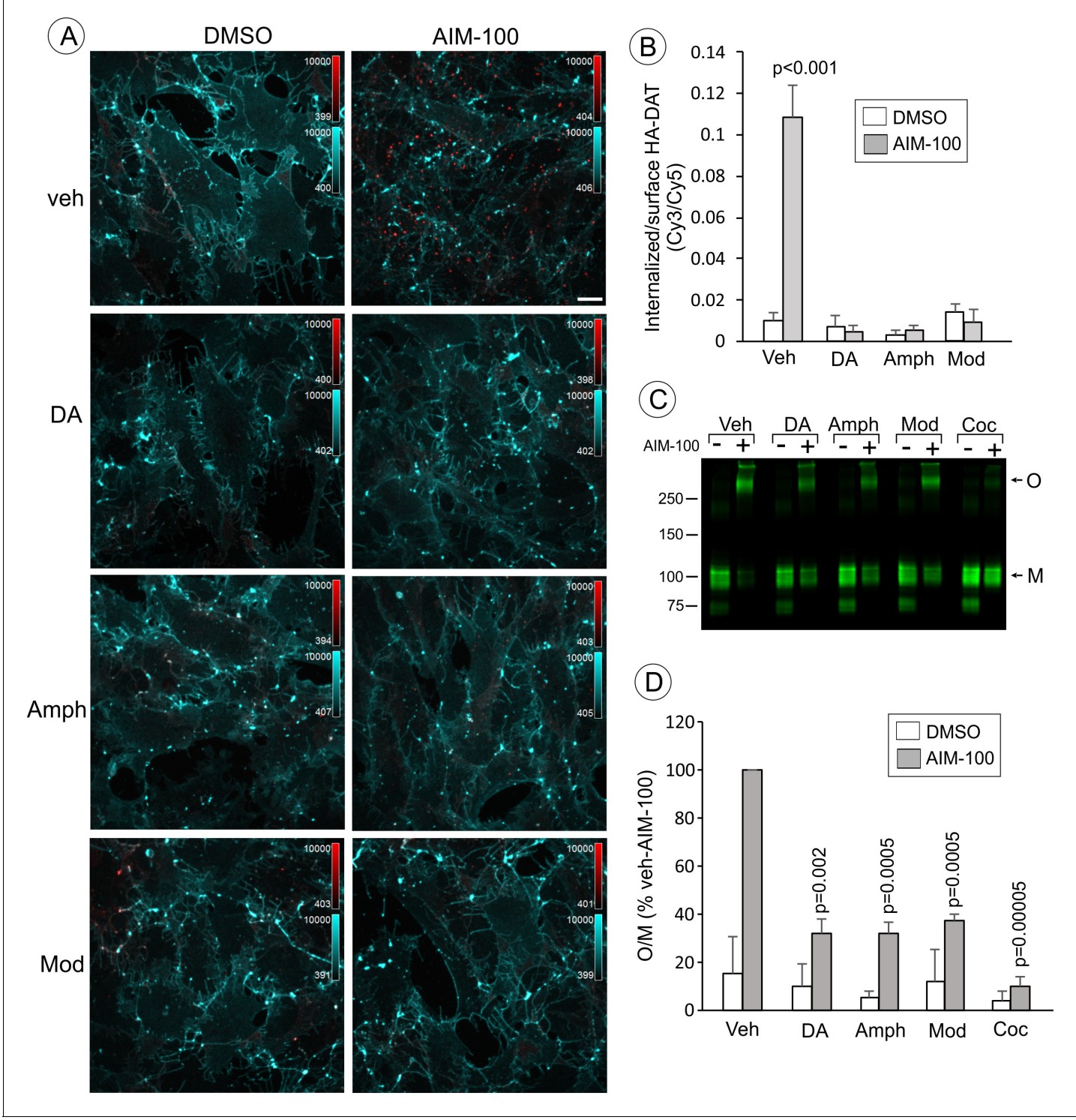

**Figure 6.** Dopamine, amphetamine and modafinil inhibit AIM-100-induced oligomerization and endocytosis of DAT. (**A**) PAE/YFP-HA-DAT cells were incubated with HA11 for 30 min at 37°C, and then incubated with vehicle (water), 100 μM dopamine (DA), 100 μM amphetamine (Amph) or 20 μM modafinil (Mod) for 15 min. The cells were further incubated with vehicle (DMSO) or 20 μM AIM-100 for 2 hr at 37°C in the same media. After fixation, cultures were stained with secondary anti-mouse antibodies conjugated with Cy5 (*surface HA-DAT*), permeabilized with Triton X-100 and stained with secondary anti-mouse conjugated with Cy3 (*internalized HA-DAT*). 3D images were acquired through 488 (YFP, not shown), 561 (Cy3, *red*) and 640 nm (Cy5, *cyan*) channels. Maximum intensity projection images are presented. Scale bar. 10 μm. (**B**) Cy3/Cy5 ratios were calculated in experiments exemplified in (**B**). Results are presented as mean values of the ratio (±S.D.; n = 4–6). P values are calculated for 'AIM-100' against other experimental variants treated with AIM-100 and inhibitors/substrates. (**C**) PAE/YFP-HA-DAT cells were incubated with vehicle, 100 μM DA, 100 μM Amph, 20 μM Mod

*Figure 6 continued on next page*

*Figure 6 continued*

or 10 µM cocaine (Coc) for 10 min. The cells were further incubated with DMSO or 20 µM AIM-100 for 2 hr at 37°C in the same media, and lysates were analyzed by immunoblotting with the GFP antibody. Representative experiment is shown. *M*, monomers; *O*, oligomers. (**D**) The mean values of the O/M ratio (±S.D.) were calculated in three independent experiments exemplified in (**C**), and expressed as percent of the O/M value determined in cells incubated with AIM-100 but not with inhibitors or substrates ('*veh-AIM-100*'). P values are calculated for cells treated with AIM-100 and inhibitors/substrates versus 'veh-AIM-100'.

DOI: https://doi.org/10.7554/eLife.32293.012

constitutive internalization in the presence of monensin was found to be significantly smaller than in the presence of AIM-100, and the monensin-dependent accumulation was not inhibited by cocaine (*Figure 9—figure supplement 2*). Therefore, AIM-100-induced accumulation of DAT in endosomes is the result of a combination of an ineffective recycling and an accelerated endocytosis, which is regulated by the mechanism apparently different from that of the constitutive endocytosis revealed in the presence of monensin.

## Mechanisms of AIM-100-induced DAT trafficking

AIM-100-dependent accumulation of DAT in endosomes of PAE (*Figure 1B*) was undetectable by fluorescence microscopy at concentrations of AIM-100 sufficient to inhibit Ack1 activity (<2 µM) (*Mahajan et al., 2010*). Therefore, we hypothesized that AIM-100-induced DAT endocytosis does not involve Ack1 under conditions of our experiments. To directly test for a specific role for Ack1, this protein was depleted by siRNA. PAE/YFP-HA-DAT cells were incubated with AIM-100 in the presence of transferrin conjugated with TexasRed (Tfn-TxR), an endocytic cargo that is internalized through the clathrin-mediated pathway. siRNA knock-down of Ack1 (65–80% depletion, *Figure 10A*) did not lead to YFP-HA-DAT endocytosis or other visible changes in the pattern of YFP-HA-DAT localization when compared with the control cells (*Figure 10B*). The amount of Tfn-TxR in endosomes was not increased but rather reduced by AIM-100 treatment or Ack1 depletion (*Figure 10—figure supplement 1*), possibly due to the involvement of Ack1 in endosomal sorting (*Grøvdal et al., 2008*). Furthermore, AIM-100 caused robust YFP-HA-DAT endocytosis in Ack1-depleted cells (*Figure 10C*) as evident by the strong increase in YFP-HA-DAT co-localization with Tfn-TxR in endosomes (*Figure 10D*). Finally, another highly efficient inhibitor of Ack1 (KRCA-0008, $IC_{50}$ = 18 nM) (*Park et al., 2013*) did not induce YFP-HA-DAT oligomerization (*Figure 10—figure supplement 2A*) and endocytosis at concentrations of 2–20 µM (*Figure 10F*). Thus, together the data in *Figure 10* indicate that AIM-100 induced DAT endocytosis is Ack1-independent in PAE cells.

The demonstration that endocytic trafficking induced by AIM-100 is not mediated by Ack1 in our experiments prompted testing the potential role of other endocytic mechanisms. Knock-down of clathrin heavy chain (CHC) had no inhibitory effect on AIM-100 induced YFP-HA-DAT endocytosis (*Figure 10A–C*) and oligomerization (*Figure 10—figure supplement 2B*), whereas Tfn-TxR endocytosis was strongly inhibited as would be expected (*Figure 10B*, insets; and *Figure 10—figure supplement 1*). On the other hand, CHC knockdown caused a partial inhibition of the constitutive endocytosis of DAT observed in the presence of monensin (*Figure 10—figure supplement 3*) [also see (*Sorkina et al., 2005*)]. These data suggest that AIM-100 re-directs DAT to a specific endocytosis pathway rather than abolishing the plasma membrane retention of DAT.

Stemming from the demonstration that AIM-100-induced endocytosis is clathrin-independent, we tested the importance of several mechanisms known to be involved in CIE. Surprisingly, AIM-100-induced endocytosis of YFP-HA-DAT was not significantly inhibited by dynamin-2 siRNA knock-down (*Figure 11A*) and overexpression of the K44A dominant-negative dynamin mutant (data not shown). Latranculin B (inhibitor of cortical actin cytoskeleton), ML141 (cdc42 inhibitor) and nystatin (cholesterol-chelating drug) also did not affect AIM-100-dependent DAT endocytosis (*Figure 11B*). Further, the PKC inhibitor, Go6976, did not block AIM-100-induced DAT endocytosis (*Figure 10—figure supplement 1*). ML141 treatment induced a moderate endocytosis of YFP-HA-DAT (*Figure 11B*), suggesting that cdc42 could be involved in DAT retention at the cell surface. Interestingly, an inhibitor of phospholipase D, FIPI reduced AIM-100-induced endocytosis two-fold. The involvement of phospholipase D may be attributed to the importance of the product of phospholipase D activity, phosphatidic acid, in membrane fission and fusion processes (*Donaldson, 2009*). Src kinase family inhibitor, PP2, and genistein, a wide-range tyrosine kinase inhibitor, did not alter the subcellular

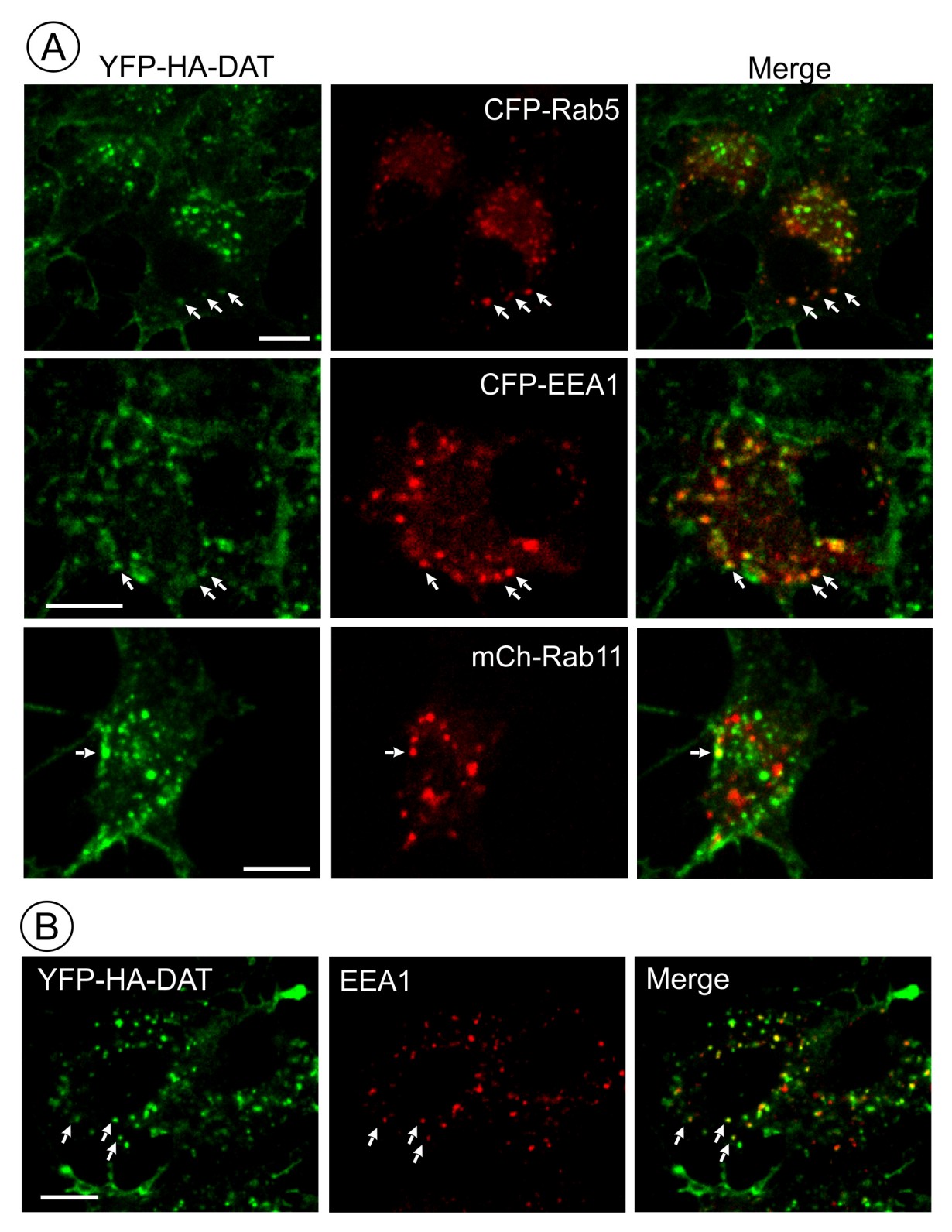

**Figure 7.** AIM-100 induced accumulation of YFP-HA-DAT in early but not recycling endosomes. (**A**) PAE/YFP-HA-DAT cells were transfected with CFP-Rab5, CFP-EEA.1, or mCherry (mCh)-Rab11, and incubated with 20 µM AIM-100 for 1.5 hr (CFP-Rab5 and CFP-EEA.1) or 2 hr (mCherry-Rab11) at 37°C. 3D images were acquired from fixed cells through 445 (CFP, *red*), 515 (YFP, *green*) and 561 nm (mCherry, *red*) channels. Individual confocal sections are presented for all except mCherry-Rab7 images where maximum intensity projection of z-stack of confocal images are presented to better demonstrate
*Figure 7 continued on next page*

*Figure 7 continued*

low extent of colocalization. (**B**) PAE/YFP-HA-DAT cells were incubated with 20 µM AIM-100 for 1.5 hr at 37°C, fixed and stained with the EEA.1 antibody followed by Cy3-conjugated secondary. 3D images were acquired through 515 (YFP, *green*) and 561 nm (Cy3, *red*) channels. Individual confocal sections are presented. Scale bars, 10 µm.

DOI: https://doi.org/10.7554/eLife.32293.013

distribution of YFP-HA-DAT in untreated or AIM-100-treated PAE cells (data not shown). These experiments ruled out the involvement of many known CIE mechanisms, such as those underlying caveolae-, CLIC/GEEC-, Arf6-, Rho-family-GTPase- and endophilin-dependent pathways, as well as micropinocytosis (*Boucrot et al., 2015*; *Mayor et al., 2014*). Thus, collectively the data in *Figures 10* and *11* suggest that AIM-100 induces DAT endocytosis via a novel CIE mechanism.

## Discussion

In the present studies, we have demonstrated the remarkably robust effects of a small-molecule tyrosine kinase inhibitor, AIM-100, in promoting DAT oligomerization, clustering at the cell surface and endocytic trafficking. All AIM-100-induced effects were highly specific to DAT as the same compound did not increase oligomerization, clustering and endocytosis of SERT and NET. The itinerary of AIM-100-induced trafficking of DAT was found to be unique as it involved a novel clathrin- and dynamin-independent endocytosis mechanism followed by the retention of internalized DAT in early Rab5/EEA.1 endosomes without significant recycling and sorting of DAT to lysosomes.

To our knowledge, the strong effect of AIM-100 on DAT oligomerization is the first example of a chemically-enhanced non-covalent oligomerization of a multi-spanning TM protein. The mechanisms of DAT dimerization and oligomerization have been studied extensively but remain unclear (*Chen and Reith, 2008*; *Hastrup et al., 2001*; *Hastrup et al., 2003*; *Sorkina et al., 2003*; *Torres et al., 2003*; *Zhen et al., 2015*). Whereas the bacterial homolog of DAT LeuT was crystalized as a dimer (*Yamashita et al., 2005*), Drosophila DAT was crystallized as a monomer (*Penmatsa et al., 2013*). We have previously used FRET and co-immunoprecipitation to demonstrate constitutive DAT oligomerization in the plasma membrane and intracellular compartments (*Sorkina et al., 2003*). In the same study, we detected a small fraction of SDS-resistant oligomers using immunoblots. Our new finding that treatment of intact cells and synaptosomes with AIM-100 robustly increases the fraction of these SDS-resistant DAT oligomers suggests that such oligomers are not the result of post-solubilization aggregation of DAT. In fact, AIM-100 did not increase DAT oligomerization when it was added to cell lysates (data not shown). Mild denaturation at 37°C (instead of 95°C) did not reduce the fraction of AIM-100-induced DAT oligomers, suggesting that this oligomerization is not thermally induced (*Figure 1—figure supplement 1*). Dramatic AIM-100-stimulated oligomerization and clustering of DAT observed using TIR-FM and FRET provide further support to the view that SDS-resistant oligomers reflect DAT oligomerization in the intact living cell. It should be noted that functionally-important SDS-resistant oligomers of other proteins, such as for example G-protein coupled receptors, SNAREs and ERGIC, have been previously described (*Mascia and Langosch, 2007*; *Neve et al., 2005*; *Salahpour et al., 2003*; *Yang et al., 1999*).

Detection of a 290 kDa species of YFP-DAT/YFP-HA-DAT (*Figures 1*, *4*, *5*, *6* and *8* and supplements) and a 190 kDa species of HA-DAT (*Figure 2D*) suggests that this 'minimal' DAT oligomer may correspond to a trimer. Although DAT and other SLC6 transporters are typically thought to be dimers, it was reported that chemical cross-linking of Cys246 and Cys306 results in the formation of dimers, trimers and tetramers of DAT (*Hastrup et al., 2001*; *Hastrup et al., 2003*). The fact that similar SDS-resistant oligomeric species of NET and SERT were not detected in the absence or presence of AIM-100 implies that despite the structural similarity of catecholamine transporters and contrary to a generally accepted notion of an equivalence in their oligomerization mechanisms (*Alguel et al., 2016*), these mechanisms might actually be fundamentally different. For example, an interaction of SERT with phosphatidylinositol-4,5-bisphosphate (PIP2) was recently demonstrated to be important for the formation of stable oligomers of SERT (*Anderluh et al., 2017*). However, this mechanism may not be involved in DAT oligomerization. DAT has been shown to be oligomeric in endosomes and the endoplasmic reticulum (*Sorkina et al., 2003*; *Zhen and Reith, 2018*), where PIP2 is present at extremely low levels (*Hammond and Balla, 2015*). In the present study we observed conversion

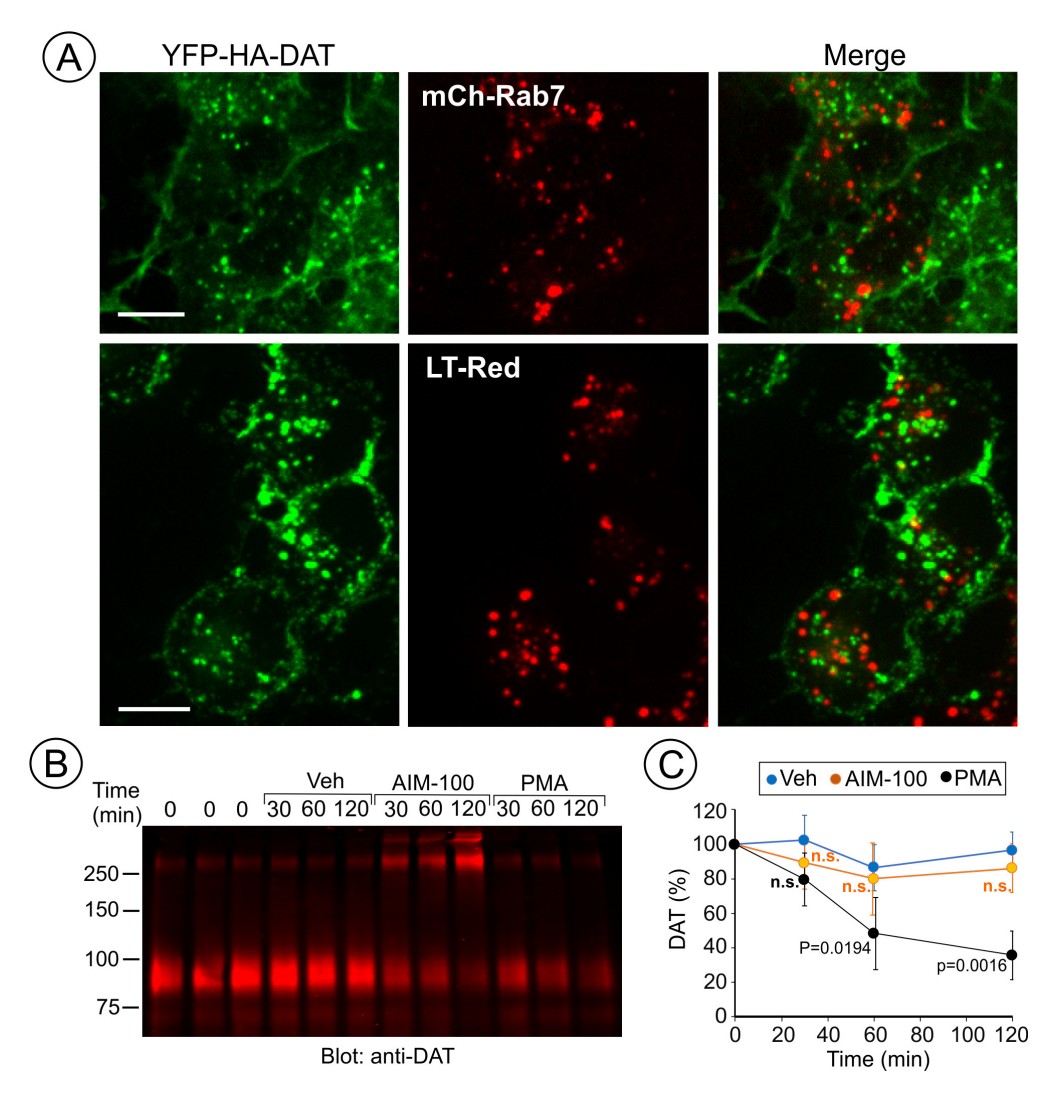

**Figure 8.** AIM-100 does not target DAT to late endosomes and lysosomes for degradation. (**A**) PAE/YFP-HA-DAT cells were transfected with mCherry-Rab7 (*mCh-Rab7*), or preincubated with LysoTrackerRed (*LT-Red*) for 5 min, and then incubated with 20 μM AIM-100 for 90 min at 37°C. 3D images were acquired through 515 (YFP, *green*) and 561 nm (mCherry or LT-Red, *red*) filter channels from fixed (mCh-Rab7) or living cells (LT-Red). Maximum intensity projections of z-stack of confocal images are presented. Scale bars, 10 μm. (**B**) PAE/YFP-HA-DAT (western blot is shown) and PAE/YFP-DAT cells were pretreated with 50 μM cycloheximide and further maintained with this inhibitor during 0–120 min incubation with vehicle, 20 μM AIM-100 or 1 μM PMA. Cell lysates were probed for DAT. (**C**) On the right, the amount of YFP-HA-DAT and YFP-DAT (sum of oligomers and monomers) was quantified. Results are presented as mean values (-/+SD, n = 3) of percent to the amount of DAT at time '0'. P values are calculated versus 'vehicle'.

DOI: https://doi.org/10.7554/eLife.32293.014

The following figure supplement is available for figure 8:

**Figure supplement 1.** AIM-100 does not induce DAT ubiquitination.

DOI: https://doi.org/10.7554/eLife.32293.015

of an monomeric immature 70 kDa species of YFP-DAT into a large oligomeric SDS-species (see *Figure 1E* and other figures). It is possible that differences in the sites of PIP2 binding to DAT and SERT (amino-terminus versus transmembrane core, respectively) (*Buchmayer et al., 2013*; *Hamilton et al., 2014*) are responsible for different mechanisms of their oligomerization.

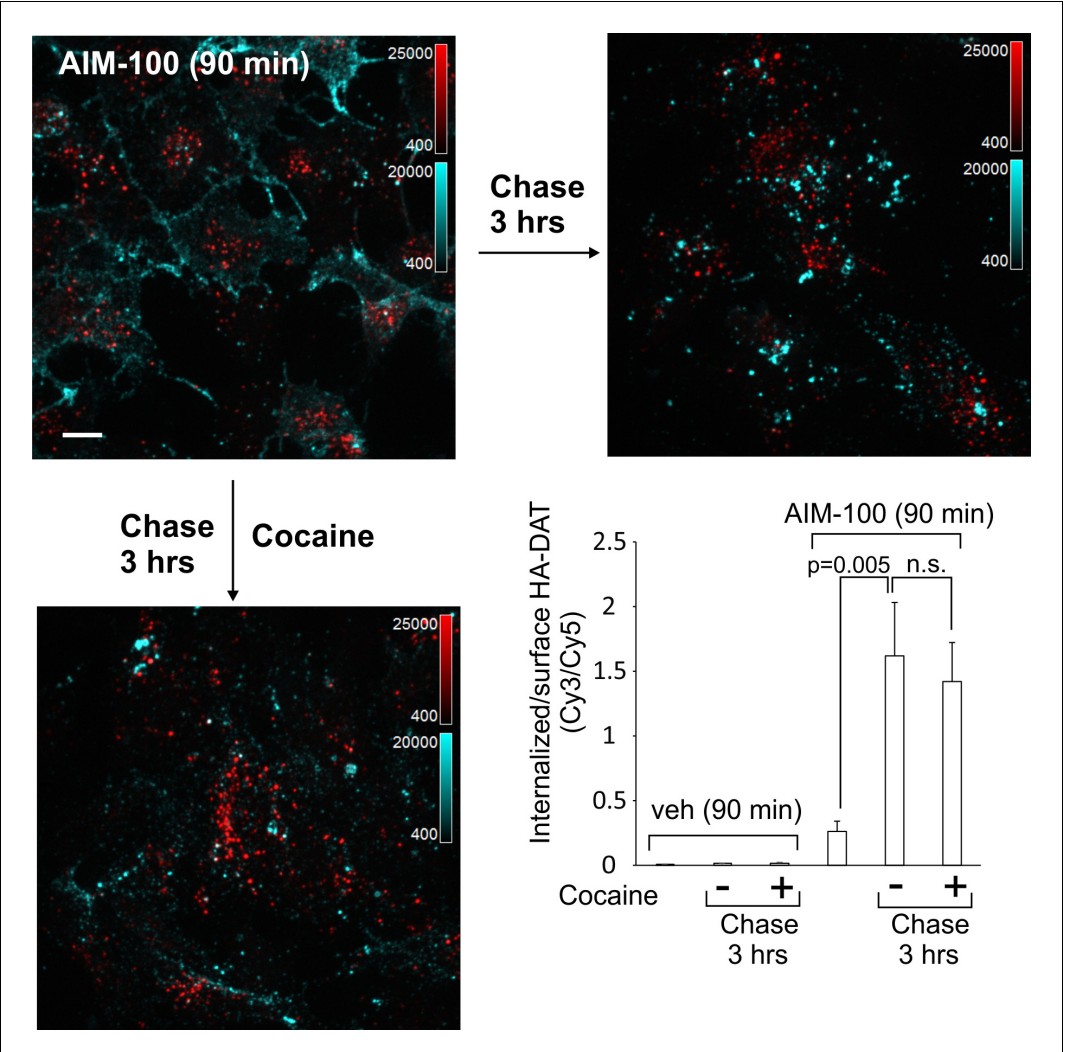

**Figure 9.** DAT internalized in AIM-100-treated cells is not recycled. PAE/YFP-HA-DAT cells were incubated with HA11 for 45 min at 37°C, washed and further incubated with vehicle or AIM-100 for 90 min at 37°C. Cells were fixed, or further incubated in F12 medium with or without 5 µM cocaine for 3 hr at 37°C before fixation. After fixation, cultures were stained with secondary anti-mouse antibodies conjugated with Cy5 (*surface HA-DAT*), permeabilized with Triton X-100 and stained with secondary anti-mouse conjugated with Cy3 (*internalized HA-DAT*). 3D images were acquired through 488 (YFP, not shown), 561 (Cy3, *red*) and 640 nm (Cy5, *cyan*) channels. Maximum intensity projections of z-stack of confocal images are presented. Scale bars, 10 µm. Cy3/Cy5 ratios were calculated, and results are presented as mean values of the ratio (±SEM; n = 7-10). n.s., p=0.19.

DOI: https://doi.org/10.7554/eLife.32293.016

The following figure supplements are available for figure 9:

**Figure supplement 1.** YFP-HA-DAT is recycled after PMA-induced internalization in the presence of AIM-100.
DOI: https://doi.org/10.7554/eLife.32293.017

**Figure supplement 2.** Endosomal accumulation of DAT in the presence of monensin is not inhibited by cocaine.
DOI: https://doi.org/10.7554/eLife.32293.018

The inhibitory action of DAT blockers and substrates on AIM-100-induced DAT oligomerization (*Figures 5–6*) suggest that AIM-100 effect may require binding of AIM-100 to a site in the DAT molecule that is inaccessible when the transporter is occupied by these substrates or inhibitors. The simplest explanation of these data is that AIM-100 acts by binding to the extracellular substrate binding sites (S1 and/or S2). However, this hypothesis is inconsistent with other data: (i) the W63A mutant is efficiently oligomerized by AIM-100 despite the fact that its substrate binding sites are inaccessible

(*Figure 5*); (ii) the substrate binding sites are accessible to AIM-100 in SERT and NET but AIM-100 does not induce their oligomerization (*Figure 4*); and (iii) AIM-100 non-competitively inhibits CFT binding to DAT (*Figure 5—figure supplement 1*). Therefore, it could be hypothesized that in addition to a low affinity binding to S1/S2 sites, AIM-100 may interact more strongly with another region of the DAT molecule but only if the DAT is in a 'flexible' folding state such that it is not locked into a specific conformation (OF, OC or IF). Such an interaction may occur at the cytoplasmic interface of the DAT molecule (AIM-100 is lipophilic and membrane-permeable), which may stabilize the DAT oligomers via an allosteric mechanism. The inability of AIM-100 to increase DAT oligomerization at 4°C (*Figure 5—figure supplement 1*) supports the hypothesis that an effective diffusion of AIM-100 across and/or within the membrane is involved in the AIM-100 action on DAT. Because the same species of the DAT oligomer was detected before and after AIM-100 treatment by immunoblotting, and because it has been shown that DAT substrates inhibit basal DAT oligomerization (*Chen and Reith, 2008*), it is possible that the molecular interfaces involved in basal/constitutive and AIM-100-induced DAT oligomerization are similar. Future studies of the structure-based mechanisms by which AIM-100 stabilizes DAT oligomers may therefore shed light on the mechanisms and functional role of the constitutive DAT oligomerization.

Formation of highly stable oligomers and DAT clustering in living cells correlated with DAT endocytosis in experiments that examined the effects of transporter inhibitors and substrates, and compared AIM-100 effects on three monoamine transporters (*Figures 4–6*). These observations together with the fast kinetics of AIM-100-induced clustering of DAT (*Figure 3B–C*) suggest that DAT oligomerization and clustering are part of the mechanism mediating AIM-100-induced DAT endocytosis. This point is supported by the observation of a higher proportion of DAT oligomers in endosomes than in the cell-surface pool of DAT molecules (*Figure 1G*). However, mutagenesis of potential oligomerization interfaces in DAT yielded misfolded DAT mutants incapable of reaching the plasma membrane (data not shown), thus precluding us to obtain a formal prove that oligomerization is necessary for endocytosis.

The extent of DAT accumulation in endosomes following treatment with AIM-100 was far greater than we have ever observed using any other stimulant of DAT endocytosis. Several lines of evidence suggest that such a dramatic accumulation is a synergistic effect due to both accelerated endocytosis and inefficient recycling (*Figures 7–9*). We propose that in early endosomes DAT oligomers are not efficiently packaged into tubular recycling carriers due to the large size of these oligomers. When monomeric receptors that normally recycle are crosslinked by multivalent ligands, these receptors have been shown to be partially mis-targeted to the lysosomal pathway (*Geuze et al., 1987*; *Hopkins and Trowbridge, 1983*). Indeed, the inability of cargo to recycle typically results in its lysosomal degradation. However, surprisingly AIM-100 did not lead to DAT degradation (*Figure 8*). It is possible that targeting of non-ubiquitinated cargo to lysosomes occurs on a substantially longer time scale than our 2 hr time-course experiments (*Tewari et al., 2015*).

Multiple lines of experimental evidence are presented in *Figure 10* to demonstrate that the AIM-100 effects are off-target and unrelated to inhibition of the Ack1 activity. This raises a question of whether the pathways and mechanisms of DAT endocytosis in the presence of AIM-100 observed by Wu and co-workers (*Wu et al., 2015*) and in the present study are different. The observation that AIM-100 did not induce endocytosis of SERT is indicative of the commonality of mechanisms analyzed in both studies. In contrast to observations by Wu and co-workers, AIM-100-induced DAT endocytosis was not affected by cdc42 inhibition and was clathrin-independent in our experiments (*Figures 10* and *11B*). The clathrin-dependency of AIM-100-induced endocytosis of DAT in SK-N-MC cells was established in the former study using a small-molecule clathrin inhibitor pistop-2 (*Wu et al., 2015*). However, considerable general toxicity of this compound has been reported (*Lemmon and Traub, 2012*; *Smith et al., 2013*; *Willox et al., 2014*). Pitstop-2 has also been shown to strongly block the CIE (*Dutta et al., 2012*). Furthermore, although Ack1 was reported to be associated with the clathrin coated structures, the importance of Ack1 for CME has not been established (*Shen et al., 2011*). Certainly, differences in the requirement for Ack1 may be attributed to different cell types, DAT expression levels or sensitivity of the endocytosis assays. Finally, it should be noted that screening for kinases involved in DAT regulation did not reveal the involvement of Ack1 (*Vuorenpää et al., 2016a*).

To conclude, we put forward a hypothetical model of AIM-100-induced DAT oligomerization and endocytosis summarized as follows. First, we propose that AIM-100 increases DAT oligomerization

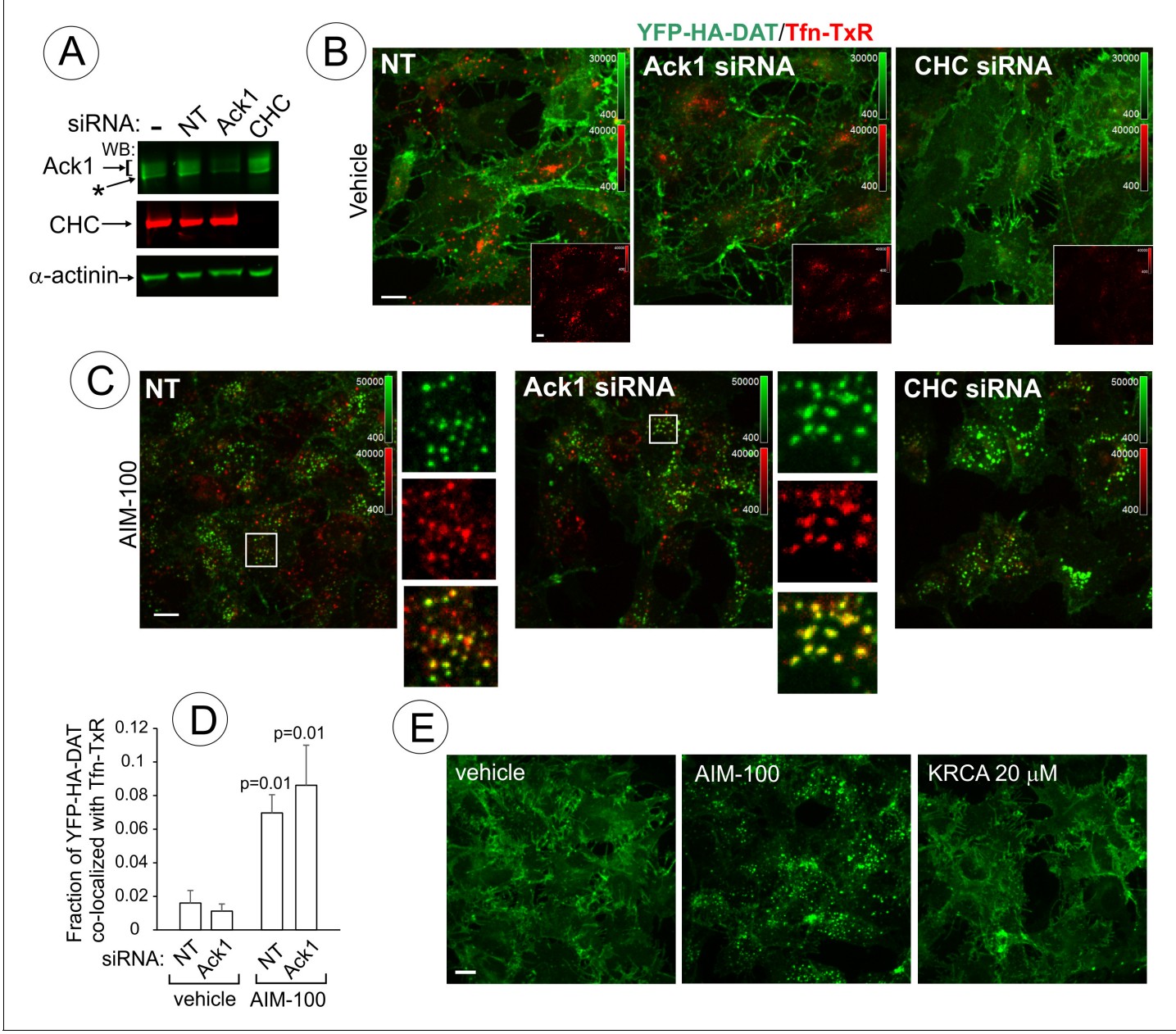

**Figure 10.** AIM-100-induced DAT endocytosis is independent of Ack1 and clathrin. (**A-D**) PAE/YFP-HA-DAT cells were transfected twice with non-targeting (NT), clathrin heavy chain (CHC) or Ack1 siRNAs. After 3–5 days, the cells were lysed and tested for the efficiency of knock-downs (**A**) or used for microscopy imaging (**B-D**). Asterisk in (**A**) indicates non-specific band recognized by Ack1 antibodies. (**B**) Cells were incubated with vehicle (DMSO) for 2 hr at 37°C in the presence of 5 µg/ml Tfn-TxR. 3D images were acquired through 488 nm (YFP) and 561 nm (TxR) channels. Merged images of maximal intensity projections of 3D images are shown. Insets show corresponding Tfn-TxR images to better demonstrate inhibition of Tfn-TxR internalization in CHC-depleted cells. Corresponding full-size images are presented in *Figure 10—figure supplement 1A*. (**C**) Cells incubated with 20 µM AIM-100 for 2 hr at 37°C in the presence of 5 µg/ml Tfn-TxR, were imaged as in (**B**). Merged images of YFP-HA-DAT and Tfn-TxR are presented. Insets show high magnification of the regions marked by white rectangle to demonstrate co-localization of Tfn-TxR and YFP-HA-DAT. (**D**) Quantification of co-localization of YFP-HA-DAT with Tfn-TxR in experiments represented in (**B**) and (**C**). Results are presented as mean values of the fraction of YFP-HA-DAT co-localized with Tfn-TxR (±SD; n = 3-7). P values are 'AIM-100' compared to 'vehicle'. (**E**) Cells were incubated with 20 µM AIM-100, vehicle (DMSO) or 20 µM KRCA-0008 for 2 hr at 37°C, fixed, and 3D images of YFP were acquired. Maximal intensity projection images are presented. Scale bars, 10 µm.

DOI: https://doi.org/10.7554/eLife.32293.019

The following figure supplements are available for figure 10:

**Figure supplement 1.** Endocytosis of Tfn-TxR in the presence of AIM-100 in cells depleted of Ack1 or CHC.

*Figure 10 continued on next page*

*Figure 10 continued*

DOI: https://doi.org/10.7554/eLife.32293.020

**Figure supplement 2.** AIM-100-induced DAT oligomerization is not induced by KRCA-0008 and not affected by CHC knockdown.

DOI: https://doi.org/10.7554/eLife.32293.021

**Figure supplement 3.** Endosomal accumulation of DAT in the presence of monensin is partially clathrin-dependent.

DOI: https://doi.org/10.7554/eLife.32293.022

by directly interacting with DAT in its conformation favorable for oligomerization and thus stabilizing DAT trimers/oligomers. We also suggest that the AIM-100:DAT interaction and the occupancy of the substrate binding sites are mutually exclusive. Second, we propose that DAT oligomerization results in an extensive surface clustering, which may be driven by the hydrophobic mismatch of oligomers and the membrane. Clustering is typically observed as the consequence of ligand-induced dimerization of receptors [for example (*Zidovetzki et al., 1981*; *Zidovetzki et al., 1986*). Third, DAT clusters may stabilize or promote spontaneous membrane invagination and scission leading to a coat-independent endocytosis. Such TM cargo-induced membrane remodeling has been proposed (*Johannes and Mayor, 2010*; *Johannes et al., 2014*), although experimentally-proven examples in intact cells or in vitro are rare (*Callenberg et al., 2012*; *Davies et al., 2012*; *Rosholm et al., 2017*; *Spoden et al., 2008*). Independence of the AIM-100-induced endocytosis of clathrin, dynamin and other mechanisms mediating various known CIE pathways, supports the hypothesis that this endocytosis may not involve a specific membrane bending/scission machinery. Thus, experiments with AIM-100 uncover a novel mechanism of endocytosis triggered by the conformation-coupled cargo oligomerization. Future studies will address whether such mechanism is utilized by other oligomeric TM proteins.

Our findings inevitably lead to a hypothesis that coat-free endocytosis may serve as part of the regulatory and quality-control mechanisms that remove excess DAT from the cell surface in the absence of extracellular substrates and may clear DAT oligomers as they are functionally impaired (*Zhen and Reith, 2018*). Defining the molecular mechanisms and mapping a novel interface of AIM-100 interaction with the DAT should open avenues to developing new approaches for modulating DAT activities and subcellular localization at the highest specificity.

# Materials and methods

**Key resources table**

| Reagent type (species) or resource | Designation | Source or reference | Identifiers | Additional information |
|---|---|---|---|---|
| Dopamine trasporter (Homo sapiens) | human DAT | n/a | Q01959 | |
| Dopamine transporter (Mus musculus) | mouse DAT | n/a | Q61327 | |
| Genetic reagent | Mouse transgenic knock-in HA-DAT | *Rao et al., 2012* (available from Jackson lab) | B6.Cg-Sklca3$^{tm1.1Asor}$/J; Stock #029381 | |
| Recombinant DNA reagent | YFP-HA-DAT | *Sorkina et al., 2006* | Addgene #90244 | |
| Recombinant DNA reagent | W63A/YFP-HA-DAT | *Sorkina et al., 2009* | Addgene #90946 | |
| Recombinant DNA reagent | YFP-DAT | *Sorkina et al., 2003* | Addgene #90228 | |
| Antibody | HA11 (mouse monoclonal) | BioLegend | clone 16B12 (MMS101P) RRID:AB_291261 | |
| Antibody | DAT (rat monoclonal) | EMD Millipore | MAB369; RRID:AB_2190413 | |
| Antibody | ubiquitin (rabbit polyclonal) | Sigma-Aldrich | U5379; RRID:AB_477667 | |
| Antibody | EEA1 (mouse monoclonal) | BD Biosciences | BDB610457; RRID:AB_397830 | |

*Continued on next page*

*Continued*

| Reagent type (species) or resource | Designation | Source or reference | Identifiers | Additional information |
|---|---|---|---|---|
| Antibody | CHC (rabbit polyclonal) | Abcam | ab21679; RRID:AB_2083165 | |
| Antibody | GFP (rabbit polyclonal) | Abcam | ab290; RRID:AB_303395 | |
| Antibody | Ack1 (mouse monoclonal) | Santa Cruz | A-11; sc-28336; RRID:AB_626629 | |
| Antibody | dynamin-2 (rabbit polyclonal) | ABR ((Thermo Fisher Scientific)) | Cat # PA1-661; RRID:AB_2293040 | |
| Antibody | GFP (mouse monoclonal) | Clontech | JL8; Cat# 632380; RRID:AB_10013427 | |
| Antibody | Cy5 (donkey-anti-mouse) | Jackson Immuno Research | 715-175-151; RRID:AB_2340820) | 1:50 in antibody uptake assay |
| Antibody | Cy3 (donkey anti-mouse) | Jackson Immuno Research | 715-165-151; RRID:AB_2315777 | 1:500 in antibody uptake assay |
| Antibody | Alexa488 (donkey anti-rat) | Jackson Immuno Research | 712-545-153; RRID:AB_2340684 | 1:500 |
| Antibody | IRDye-800 (goat anti mouse) | LI-COR | 926–32210; RRID:AB_621842 | 1:20000 |
| Antibody | IRDye-680 (goat anti mouse) | LI-COR | 926–32220; RRID:AB_621840 | 1:20000 |
| Chemical compound, drug | Tfn-TxR (transferrin-Texas Red) | Invitrogen | Cat #T2875 | |
| Chemical compound, drug | LT-Red (LysoTrackerRed) | Invitrogen | Cat# L7528 | |
| Chemical compound, drug | AIM-100 | EMD MilliporeSigma | Cat#104833 | |
| Chemical compound, drug | KRCA-0008 | EMD MilliporeSigma | Cat# SML1304 | |
| Chemical compound, drug | PMA | EMD MilliporeSigma | Cat# P8139 | |
| Chemical compound, drug | Go6976 | EMD MilliporeSigma | Cat# 365250 | |
| Chemical compound, drug | Latranculin B | EMD MilliporeSigma | 428020 | |
| Transfected construct | siRNA to Ack1 | Integrated DNA Technologies Inc. | Cat# 87793753; 87793756 | |
| Transfected construct | siRNA to CHC | Thermo Fisher Scientific | *Sorkina et al., 2005* | |
| Cell line (human) | SH-SY5Y | ATCC | RRID:CVCL_0019 | |
| Cell line (human) | HEK293T | ATCC | RRID:CVCL_0063 | |
| Cell line (human) | HeLa | ATCC | RRID:CVCL_0030 | |
| Cell line (porcine) | PAE | *Westermark et al., 1990* | PMID: 2153283 | |
| Software | SlideBook6 | Intelligent Imaging Innovations, Inc. | | |
| Software | Odyssey Application Software 3.0 | Li-COR, Inc. | | |

## Antibodies and chemicals

Antibodies were purchased from the following sources: monoclonal rat antibody against the N-terminus of DAT (MAB369) from EMD Millipore (Bellerica, MA); rabbit polyclonal antibodies to ubiquitin and actin from Sigma-Aldrich (St. Louis, MO); mouse monoclonal antibody to hemagglutinin epitope HA11 (16B12) from BioLegend (Dedham, MA); monoclonal mouse antibody to EEA.1 from BD Biosciences (Franklin Lakes, NJ); polyclonal rabbit antibodies to CHC and GFP from Abcam (Cambridge, MA), mouse monoclonal antibody to Ack1 from Santa Cruz (Dallas, TX); mouse monoclonal antibody to GFP from Clontech (Mountain View, CA); rabbit polyclonal antibody to α-actinin from Cell Signaling Technologies (Danvers, MA); rabbit polyclonal antibody to dynamin-2 were from ABR (now Invitrogen). Secondary donkey anti-mouse, anti-rat and anti-rabbit AffiniPure antibodies conjugated with Alexa488, Cy5, and Cy3 from Jackson Immuno Research (West Grove, PA); IRDye-800 and IRDye-680-conjugated goat anti-mouse, anti-mouse IgG1, anti-mouse IgG2a, anti-rat and

anti-rabbit antibodies were purchased from LI-COR Biosciences (Lincoln, NE). Transferrin-Texas Red, Lysotracker Red and Protein G-Sepharose were purchased from Invitrogen (Grand Island, NY). Paraformaldehyde was from Electron Microscopy Sciences (Hatfield, PA). Tissue culture reagents were purchased from Gibco, Thermo Fisher Scientific (Waltham, MA). Dopamine, amphetamine, modafinil, (2)−2-β-carbomethoxy-3-β-(4-fluorophenyl)tropane (CFT) naphthalenedisulfonate monohydrate, AIM-100, KRCA-0008, phorbol 12-myristate 13-acetate (PMA), Gö6976, Triton X-100, protease inhibitors and most other chemicals were purchased from EMD MilliporeSigma (Bellerica, MA). AIM-100 was stored as 20 mM stock solution in DMSO and diluted in the medium to a final concentration of 1–20 µM by vigorous shaking or pipetting immediately prior to incubation with cells. [3-H]CFT (82.6 Ci/mmol) was from PerkinElmers.

## DNA constructs

mCherry- or CFP-tagged Rab5, Rab11, EEA,1 and Rab7 were described previously (*Galperin and Sorkin, 2003*; *Galperin and Sorkin, 2008*; *Galperin et al., 2004*). GFP-fused human NET (GFP-NET) construct was described previously (*Paczkowski and Bryan-Lluka, 2004*). GFP-SERT was constructed by inserting human SERT cDNA between the KpnI and XbaI sites of the EGFP-C1 vector.

## Cell culture and transfections

PAE cells were originally obtained from University of Uppsala (Sweden) (*Westermark et al., 1990*). PAE cells stably expressing YFP-HA-DAT or YFP-DAT were described in several our studies [for example (*Sorkina et al., 2003*; *Sorkina et al., 2006*)]. PAE cells stably expressing the W63A mutant of YFP-HA-DAT (clonal pool) were described in (*Sorkina et al., 2009*). Human neuroblastoma cells SH-SY5Y were purchased from ATTC. SH-S5Y5 cells expressing human YFP-HA-DAT were generated by single-cell cloning in the presence of G418 (400 µg/ml). Clonal pools of PAE cells stably expressing GFP-SERT or GFP-NET were also generated by G418 selection. PAE cells were grown in F12 medium with 10% fetal bovine serum (FBS). SH-S5Y5 cells were grown in DMEM/F12 containing 10% FBS. HEK293 cells stably expressing untagged DAT and HeLa cells stably expressing YFP-HA-DAT (*Sorkina et al., 2013*; *Sorkina et al., 2006*) were grown in DMEM containing 10% FBS. All human cell lines were confirmed by genotyping and regularly checked for mycoplasma using Lonza (Allendale, NJ) detection kit.

The cells were grown on 35 mm glass-bottom dishes (MatTek, Ashland, MA) or glass coverslips for live-cell imaging experiments; on glass coverslips for direct imaging of YFP and immunofluorescence in fixed cells, or 12-well tissue culture plates for biochemical experiments.

DNA constructs were transfected using Effectine kit (Qiagen, Valencia, CA). siRNAs to Ack1 was purchased from Integrated DNA Technologies Inc (Coralville, IA); siRNAs to CHC, dynamin-2 and non-targeting siRNA were described previously (*Sorkina et al., 2013*; *Sorkina et al., 2005*). siRNA transfections were performed with DharmaFECT Transfection Reagent #1 from Thermo Fisher Scientific according to manufacturer's recommendations. Typically, transfection with siRNAs was repeated after two days, pooled cells from each control or experimental knockdown were plated for all experimental conditions on the second day after the second transfection, and experiments were performed on the third day after the second transfection. In some single transfection experiments with siRNA to CHC or dynamin-2 was used, and the cells were assayed on the third day after transfection. The efficiency of target protein knock-down was determined for each experiment by Western blotting.

## DA neuronal cultures

All experimental procedures were performed in strict accordance with the recommendations in the Guide for the Care and Use of Laboratory Animals of the National Institutes of Health. All of the animals were handled according to the approved Institutional Animal Care and Use Committee (IACUC) protocol (#16088832) of the University of Pittsburgh. Primary mesencephalic postnatal cultures were prepared from HA-DAT knock-in mice as previously described (35). Experiments were performed on primary cultures grown on glass coverslips at days in vitro (DIV) 6–10 as described (*Rao et al., 2012*).

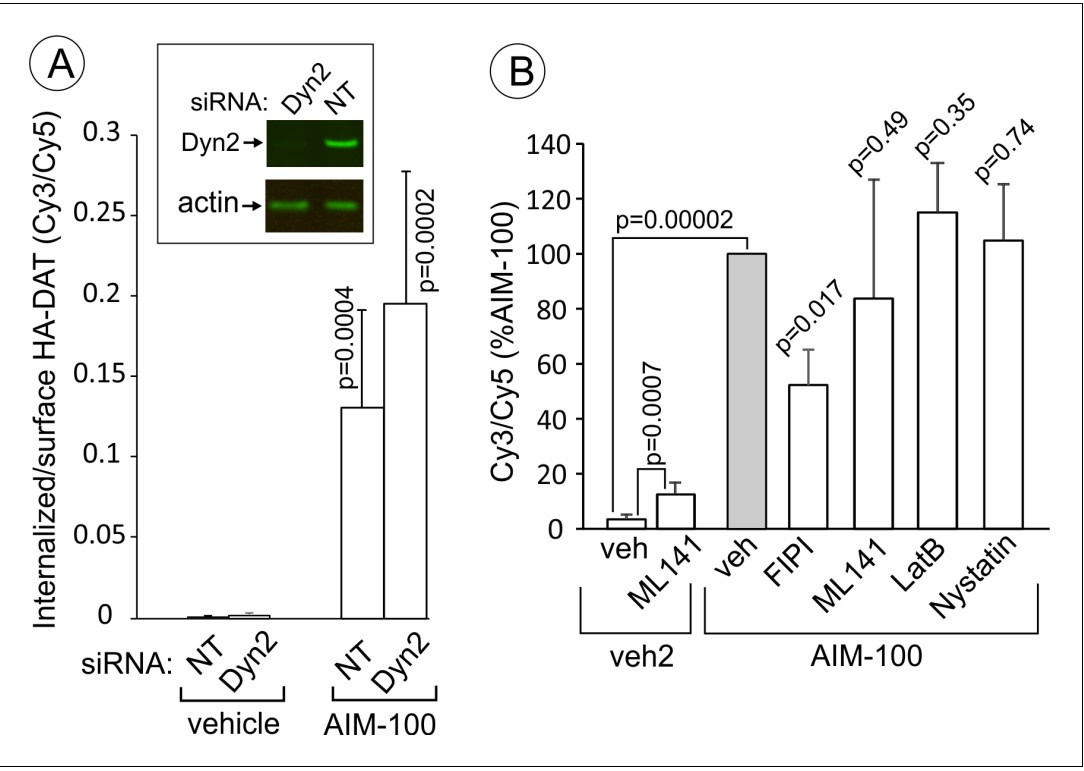

**Figure 11.** AIM-100-induced DAT endocytosis is independent on dynamin-2, actin cytoskeleton, cdc42 and cholesterol-rich lipid rafts. (**A**) PAE/YFP-HA-DAT cells were transfected twice with non-targeting (NT) or dynamin-2 (Dyn2) siRNAs. After 3–5 days, the cells were lysed and tested for the efficiency of knock-downs or for microscopy imaging. Cells were incubated with HA11 for 45 min at 37°C, and then incubated with (vehicle) or 20 µM AIM-100 for 2 hr at 37°C. After fixation, cultures were stained with secondary anti-mouse antibodies conjugated with Cy5 (*surface HA-DAT*), permeabilized with Triton X-100 and stained with secondary anti-mouse conjugated with Cy3 (*internalized HA-DAT*). 3D images were acquired through 488 (YFP), 561 (Cy3) and 640 nm (Cy5) channels. Cy3/Cy5 ratios were calculated, and the results are presented as mean values of the ratio (±SD, n = 5-6). P values are for 'AIM-100' compared to 'vehicle'. (**B**) Cells were incubated with HA11 for 1 hr at 37°C, and then incubated with vehicles corresponding to ML141 (10 µM), FIPI (1 µM), Latranculin B (LatB, 0.4 µM) or nystatin (25 µM) for 15 min at 37°C. Cells were then incubated with (DMSO; vehicle-2) or 20 µM AIM-100 for 2 hr at 37°C in the presence of inhibitors. After fixation, cultures were stained and imaged as in (**A**). Cy3/Cy5 ratios were calculated, and the results are presented as mean values of the ratio (±SD, n = 6) relative to this value in vehicle-pretreated/AIM-100-treated cells (grey bar). P values are for 'AIM-100 plus an inhibitor' versus 'AIM-100 plus vehicle', unless indicated otherwise on the graph.

DOI: https://doi.org/10.7554/eLife.32293.023

## Antibody uptake endocytosis assay and immunofluorescence

The endocytosis assay using HA11 antibody was previously described (*Sorkina et al., 2013*). Briefly, the cells grown on glass coverslips were incubated with 2 µg/ml HA11 in conditioned media (same media the cells were grown) for 30–60 min at 37°C, and then in F12 medium with various drugs at 37°C for the indicated times. The cells were washed with ice-cold Hanks balanced salt solution (HBSS) (Thermo Fisher Scientific) and fixed with freshly prepared 4% paraformaldehyde for 15 min at room temperature. The cells were washed in Dulbecco phosphate buffer saline (DPBS) (Thermo Fisher Scientific) and DPBS containing 0.5% BSA (DPBS/BSA). To occupy surface-DAT bound HA11, cells were incubated for 1 hr with 5 µg/ml secondary donkey anti-mouse antibody conjugated with Cy5 in DPBS/BSA. After triple wash and additional 15 min fixation, the cells were permeabilized with 0.1% Triton X-100 in DPBS/BSA for 5 min, and then incubated with 0.5 µg/ml the same secondary antibody conjugated with Cy3 in DPBS/BSA for 45 min to label internalized HA11. All antibody incubations were at room temperature, and followed by a 2 min triple wash in DPBS/BSA. Both primary

and secondary antibody solutions were precleared by centrifugation at 100,000 x g for 20 min. Coverslips were mounted on slides in Mowiol (Calbiochem, La Jolla, CA).

For conventional immunofluorescence staining, the cells on coverslips were fixed with paraformaldehyde and permeabilized with Triton X-100 as above, incubated with appropriate primary and secondary antibodies, each followed by triple washes, and mounted in Mowiol.

## Fluorescence microscopy

To obtain high resolution three-dimensional (3D) images of the cells, a z-stack of confocal images was acquired using a spinning disk confocal imaging system based on a Zeiss Axio Observer Z1 inverted fluorescence microscope (with 63x Plan Apo PH NA 1.4), equipped with a computer-controlled Spherical Aberration Correction unit, Yokogawa CSU-X1, Vector photomanipulation module, Photometrics Evolve 16-bit EMCCD and Hamamatsu Orca-Flash4.0 CMOS cameras, environmental chamber and piezo stage controller and lasers (405, 445, 488, 515, 561, and 640 nm), all controlled by SlideBook six software (Intelligent Imaging Innovation, Denver, CO). Typically, 15–30 serial two-dimensional confocal images were recorded at 200–400 nm intervals. All image acquisition settings were identical for all experimental variants in each experiment.

## Image analysis

To quantify of the relative amount of Cy5 (surface) and Cy3 (internalized) fluorescence in images obtained in HA11 antibody endocytosis assay, background-subtracted 3D images were segmented using minimal intensities of Cy5 (non-permeabilized cells staining) and Cy3 (permeabilized cells staining) as low thresholds to obtain Masks #1 and #2 corresponding to the total amount of surface and intracellular HA11, respectively. Additionally, a segment mask #3 of Cy3 fluorescence overlapping with Cy5 positive pixels was generated to determine the amount of Cy3-labeled antibodies that bind to surface HA11 due to incomplete occupancy of the surface HA11 with Cy5-labeled secondary antibodies before cell permeabilization. Mask #3 was subtracted from Mask #2 to obtain Mask #4 corresponding to the corrected Cy3 fluorescence (internalized HA11). The integrated voxel intensity of Masks #1 and #4 (in arbitrary linear units of fluorescence intensity; a.l.u.f.i.) were quantitated in each image containing typically 5–15 cells, and the ratio of Mask#4 to Mask#1 integrated intensities (Cy3/Cy5 ratio) was calculated to determine the extent of DAT endocytosis.

## Quantitation of endosomal Tfn-TxR and YFP-DAT:Tfn-TxR co-localization

Cells incubated with AIM-100 and Tfn-TxR were fixed and 3D images were acquired through 515 nm (YFP) and 561 nm (TexasRed) channels. The background-subtracted 3D images were deconvoluted and segmented using a minimal intensity of YFP and TexasRed (located in punctate structures – endosomes) to obtain segment masks #1 and #2, respectively, corresponding to the total amount of YFP and endosomal TexasRed, correspondingly. Mask #3 was generated that contained image voxels that overlapped in Masks #1 and #2 (corresponding to co-localized YFP and TexasRed). The ratio of integrated voxel intensity of Mask #3 to Mask#1 was quantitated in each image typically covering 12–15 confluent cells to determine the fraction of YFP-HA-DAT co-localized with Tfn-TxR relative to the total cellular YFP-HA-DAT. The total amount of endocytosed Tfn-TxR per an image of the confluent cell monolayer was determined using Mask#2.

## TIR-FM imaging and image analysis

PAE/YFP-HA-DAT cells in Mat-Tek dishes were imaged on a Nikon Eclipse Ti inverted microscope (Nikon, Melville, NY, USA) with a 100 × 1.49 NA oil-immersion objective using 488 nm laser line, and with 1–2 min intervals between time frames. All experiments were performed at 37°C and 5% CO2 in F12 medium. AIM-100 was added during continuous image acquisition. Images were collected using Nikon Elements software (version 4.30, Nikon, Melville, NY) and an Andor (Belfast, Ireland) Zyla 5.5 camera; at full resolution under these conditions the pixel size with a 1 × coupler matches Nyquist sampling (120 nm xy exactly).

Image analysis to generate kymographs was performed using Elements whereas calculations of the total number of YFP spots was performed using Imaris 7.1 software (Bitplane INC, South Windsor, CT). Images were thresholded, and particle tracking function was used to define and quantify

number of YFP puncta. Automatic detection of particles was performed using appropriate threshold values and confirmed by visual inspection of correct particle detection. Brownian motion particle-tracking algorithm was applied to trace objects through sequential frames and calculate the fraction of immobile spots.

### Cell-surface biotinylation

The cell-surface biotinylation was previously described (*Sorkina et al., 2013*). Briefly, cells grown in 12-well plates were incubated with vehicle or AIM-100 for 2 hr at 37°C, washed with ice-cold DPBS, and incubated with 1.5 mg/ml Sulfo-NHS-SS-biotin (Thermo Fisher Scientific) for 40 min at 4°C in DPBS and then quenched with 100 mM glycine-HCl buffer. After rinsing with DPBS, the cells were solubilized in TGH (1%Triton X-100, 10% glycerol, 20 mM HEPES, 50 mM NaCl) lysis buffer supplemented with 1% deoxycholate, 10 mM N-ethyl maleimide, iodacetamide, and protease (including MG132) and phosphatase inhibitors for 15–30 min at 4°C. Lysates were centrifuged at 16,000x g for 20 min to remove insoluble material. Aliquots of lysates were taken for input control, and a similar aliquot was incubated with NeutroAvidin-Sepharose (Thermo Fisher Scientific) at 4°C for 1 hr to pull-down biotinylated proteins. Triple-washed NeutroAvidin precipitates and aliquots of cell lysates were denatured in Laemmli buffer (2%SDS, 5% β-mercaptoethanol, 10% Glycerol, 62.5 mM TrisHCl, pH6.8) for 5 min at 95°C, resolved by 7.5% SDS-PAGE, transferred to nitrocellulose (Li-COR) and probed with appropriate primary and secondary antibodies conjugated to far-red fluorescent dyes (IRDye-680 or −800) followed by detection using Odyssey Li-COR system. Quantifications were performed using Li-COR software. Molecular mass of stained markers was verified by Coomassie Blue staining of unlabeled markers from BioRad.

### DAT degradation experiments

Cells grown in 12-well plates were pre-incubated with 50 µM cycloheximide (CHX) at 37°C for 30 min, and then incubated with vehicle, PMA or AIM-100 for 0–2 hr at 37°C. The cells were solubilized, the lysates were resolved by SDS-PAGE and processed for Western blotting as described above in biotinylation experiments.

### Striatal synaptosomes preparation

Preparation of striatal synaptosomes is described in more detail at Bio-protocol (*Ma and Sorkin, 2018*). 8 week old HA-DAT knock-in mice (*Rao et al., 2012*) of either sex were euthanized and decapitated. Brains were quickly removed and rinsed in ice-cold Gey's balanced salt solution with 10 mM D-glucose. Striatal tissue was collected from the brain and homogenized using a glass homogenizer with ice-cold 5 mM HEPES (with 0.32 M sucrose, pH 7.4, 1 ml for 45–50 mg striatal tissue from one brain). The homogenate was centrifuged at 1000xg for 10 min at 4°C, and the resulting supernatant was centrifuged at 12,500xg for 20 min at 4°C to pellet the synaptosomes. Synaptosomes were re-suspended in Krebs Ringer solution (KRH/G, 140 mM NaCl, 5 mM KCl, 2 mM $CaCl_2$, 1 mM $MgCl_2$, 5.5 mM HEPES and 10 mM D-glucose, pH 7.4) for experiments.

### DAT oligomerization analysis in synaptosomes

Synaptosomes were equilibrated in KRH/G at 4°C for 2 hr, pelleted by centrifugation at 12,500x g for 20 min at 4°C, and re-suspended in fresh KRH/G (one striatum in 1 ml) by pipetting 10 times with 1 ml pipet tip. Equal aliquots of re-suspended synaptosomes were incubated with vehicle (DMSO) or 10–20 µM AIM-100 for 2 hr at 37°C on nutator, centrifuged at 12,500x g for 20 min at 4°C, re-suspended in KRH/G and centrifuged again. Pelleted synaptosomes were denatured by heating in Laemmli buffer and processed for SDS-PAGE/Western blot analysis as described above in biotinylation experiments.

### FRET imaging and analysis

PAE cells co-expressing YFP-DAT and CFP-DAT (pool of cells expressing various levels of YFP- and CFP-DATs) were described previously (*Sorkina et al., 2003*). The cells grown on 35 mm MatTek dishes were incubated with DMSO (vehicle) or AIM-100 at 37°C. A method of sensitized FRET measurement, that has been described in our studies previously was used (*Galperin et al., 2004*; *Jiang and Sorkin, 2002*; *Sorkin et al., 2000*; *Sorkina et al., 2003*). Briefly, images were acquired

using the spinning disk confocal microscope at room temperature (to avoid artifacts due to the rapid movement of endosomes and filopodia during image acquisition) sequentially through 445 nm (CFP; excitation - 445 nm, emission - 470 nm), 515 nm (YFP; excitation - 515 nm, emission - 542 nm) and FRET (excitation at 445 nm; emission at 542 nm) filter channels. All image acquisition parameters were identical in experimental measurements in cells co-expressing CFP-DAT and YFP-DAT, and in control measurements of the bleed-through coefficients in cells expressing only CFP-DAT or YFP-DAT which were incubated with DMSO or AIM-100.

Corrected FRET (FRET$^C$) was calculated on a pixel-by-pixel basis using a FRET module of the SlideBook6 software. Bleed-through coefficients were measured in cells expressing only CFP-DAT or YFP-DAT, and treated or not with AIM-100. FRET$^C$ images are presented in a pseudocolor mode. FRET$^C$ intensity is displayed stretched between the low and high renormalization values, according to a temperature-based lookup table with blue (cold) indicating low values and red (hot) indicating high values. To eliminate the distracting data from regions outside of cells or cells that do not express both CFP and YFP, the CFP channel was used as a saturation channel, and the FRET$^C$ images were displayed as CFP intensity-modulated images. In these images, data with CFP values greater than the high threshold of the saturation channel are displayed at full saturation, whereas data values below the low threshold are displayed with no saturation (i.e. black).

FRET$^C$ values were also calculated from the mean fluorescence intensities for multiple selected sub-regions of the images containing individual ruffles and filopodia (plasma membrane) in vehicle-treated cells, and plasma membrane clusters and endosomes in AIM-100 treated cells as described (*Sorkina et al., 2003*). Normalized sensitized FRET (FRETN) values for individual subcellular structures, regions and compartments were calculated according to the equation (1): FRETN = FRET$^C$/YFP x CFP, where FRET$^C$, CFP and YFP are the mean background-subtracted intensities of FRET$^C$, CFP and YFP fluorescence in the selected sub-region. Because FRETN displays a non-linear dependence when donor or acceptor are in a significant molar excess over each other, FRETN values were calculated in the sub-regions of the cell in which the relative stoichiometry of the donor and acceptor was not more than 3. All calculations were performed using SlideBook6.

## Measurement of CFT binding parameters

Binding saturation analysis was used to determine $K_D$ and Bmax values of CTF binding to YFP-DAT as described (*Lin et al., 1999*). Binding assays were performed using 1 nM [3-H]CFT that was adjusted to 4.12, 7.25, 13.5, 26, 51, and 101 nM concentrations using unlabeled CFT. Non-specific binding was determined in the presence of 20 μM cocaine. Binding assays were carried out in 48-well plates in F12 medium for 30 min at 20°C, conditions minimizing AIM-100 induced endocytosis. The lack of detectable YFP-DAT endocytosis was confirmed in cells grown in parallel MatTek dishes by live-cell imaging. CFT binding was terminated by three washes with ice-cold medium. Cells were solubilized in 0.35 ml 1N NaOH, and the radioactivity was determined using a liquid scintillation counter. Cells from the parallel wells were solubilized in TGH and used for protein measurements. Calculations of $K_D$ and Bmax values were performed using GraphPad.

## Statistical analysis

Statistical significance (P value) was calculated using unpaired or paired two-tailed Student's t tests or two-way ANOVA (GraphPad and Excel). Data normality was analyzed using the D'Agostino–Pearson test (GraphPad).

## Acknowledgements

We thank Dr. Hongying (Mary) Cheng (University of Pittsburgh) for discussion of the data. This work was supported by NIH/NIDA grant DA014204 (AS).

## Additional information

### Funding

| Funder | Grant reference number | Author |
|---|---|---|
| National Institutes of Health | DA014204 | Alexander Sorkin |

The funders had no role in study design, data collection and interpretation, or the decision to submit the work for publication.

### Author contributions

Tatiana Sorkina, Conceptualization, Validation, Investigation, Visualization, Methodology, Writing—review and editing; Shiqi Ma, Investigation, Methodology, Writing—review and editing; Mads Breum Larsen, Resources, Investigation, Methodology; Simon C Watkins, Formal analysis, Investigation, Visualization, Methodology, Writing—review and editing; Alexander Sorkin, Conceptualization, Resources, Data curation, Formal analysis, Supervision, Funding acquisition, Validation, Investigation, Writing—original draft, Writing—review and editing

### Author ORCIDs

Alexander Sorkin http://orcid.org/0000-0002-4446-1920

### Ethics

Animal experimentation: This study was performed in strict accordance with the recommendations in the Guide for the Care and Use of Laboratory Animals of the National Institutes of Health. All of the animals were handled according to approved institutional animal care and use committee (IACUC) protocols (#16088832) of the University of Pittsburgh.

### Decision letter and Author response

Decision letter https://doi.org/10.7554/eLife.32293.026
Author response https://doi.org/10.7554/eLife.32293.027

## Additional files

### Supplementary files

• Transparent reporting form
DOI: https://doi.org/10.7554/eLife.32293.024

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
