## [Decision Letter]

Thank you for submitting your article "Small molecule induced oligomerization, clustering and clathrin-independent endocytosis of the dopamine transporter" for consideration by *eLife*. Your article has been favorably evaluated by Anna Akhmanova (Senior Editor) and three reviewers, one of whom is a member of our Board of Reviewing Editors. One of the reviewers, Harald Sitte, has agreed to reveal his identity.

The reviewers have discussed the reviews with one another and the Reviewing Editor has drafted this decision to help you prepare a revised submission.

This is an interesting and well-written report that describes how an inhibitor of the tyrosine kinase Ack1 (AIM-100) promotes strong internalization of the dopamine transporter (DAT) in a way that is independent of kinase inhibition and likely involves an allosteric mechanism that might be associated with the conformational state of DAT (possibly outward open configuration) and oligomerization. The internalization is found to be independent of dynamin, cholesterol and the actin cytoskeleton, suggestive of a novel internalization mechanism.

Summary:

The manuscript by Sorkina et al. describes how an inhibitor of the tyrosine kinase Ack1 (AIM-100) promotes strong internalization of the dopamine transporter (DAT) in a way that is independent of kinase inhibition and likely involves an allosteric mechanism that might be associated with the conformational state of DAT (possibly outward open configuration) and oligomerization. The internalization is found to be independent of dynamin, cholesterol and the actin cytoskeleton, suggestive of a novel internalization mechanism. The authors also find an unusual itinerary for the DAT internalization of DAT with accumulation of the transporter in early endosomes and little recycling and degradation. Overall, these findings reveal a novel mechanism for DAT trafficking and its oligomerization that are driven by the conformational-state of the transporter. There are however several issues that must be addressed as outlined below.

Essential revisions:

1) An important issue is the cell physiological relevance of the findings. To be critical, the authors find that a compound somehow affects DAT so it somehow oligomerizes, internalizes and accumulates in early endosomes without being degraded or recycled. The fact that the effect is conformation specific is interesting but the reviewers miss some experiments and a discussion that further deal with the physiological relevance of the observations in relation to DAT as such and in relation to the putative novel internalization mechanism. One obvious question is whether the effect is affected by the presence of substrates? Is the effect promoted or inhibited by dopamine and/or amphetamine? The authors specifically raise the issue whether or not AIM-100 binds to the substrate binding site or to another allosteric site so it seems obvious to address this question.

2) Along the same lines, because the effect of AIM-100 seems to be dependent on the inward facing conformation and the experiments are not conclusive about whether the compound binds to an allosteric site, it would be interesting to assess the effect of an inhibitor that preferentially binds an inward-facing conformation, such as ibogaine.

3) Also, does AIM-100 inhibit binding of a radiolabeled inhibitor to the transporter and, if so, is the inhibition competitive or allosteric in nature? This would further address the issue of how AIM-100 exerts its action on DAT.

4) Figure 9 should be supplemented. Do Ack knock-down/inhibition or knock-down/overexpression of K44A dominant negative affect the AIM-100 induced multimerization of DAT?

5) Sorkin and colleagues were among the pioneers to use FRET microscopy in neurotransmitter transporters and revealed for the first time that DAT forms oligomers in living cells throughout the secretory pathway and at the cell surface. Given that these authors have such a strong background in fluorescence microscopy in the living cell, it is rather surprising that this methodology is not being used in the current manuscript where oligomerization is in the main focus. Hence, it is not clear to the reviewer why only biochemical assays were used to make one of the two main statements in this paper. At the very least, it would be desirable to see that AIM-100 is enhancing the oligomeric properties of DAT also under more physiological conditions, i.e. in living cells.

6) The authors argue that an important control was unaffected by AIM-100: "Furthermore, AIM-100 did not inhibit recycling of the transferrin receptor (data not shown)." Here, my question is whether AIM-100 induces internalization of the TFR at all (which would also be a nice and independent control) and furthermore, to show these data.

---

## [Author Response]

Essential revisions:

1) An important issue is the cell physiological relevance of the findings. To be critical, the authors find that a compound somehow affects DAT so it somehow oligomerizes, internalizes and accumulates in early endosomes without being degraded or recycled. The fact that the effect is conformation specific is interesting but the reviewers miss some experiments and a discussion that further deal with the physiological relevance of the observations in relation to DAT as such and in relation to the putative novel internalization mechanism. One obvious question is whether the effect is affected by the presence of substrates? Is the effect promoted or inhibited by dopamine and/or amphetamine? The authors specifically raise the issue whether or not AIM-100 binds to the substrate binding site or to another allosteric site so it seems obvious to address this question.

We performed a series of experiments to test the effects of dopamine and amphetamine. These experiments demonstrated that high concentrations of substrates (20 μM and higher) significantly inhibit AIM-100-induced DAT oligomerization and endocytosis. These data are summarized in new Figure 6. Please see the second paragraph of the subsection “DAT inhibitors and substrates diminish the effects of AIM-100 on DAT”, and the figure legend for description of these experiments.

We discuss the physiological relevance of our observations in the “Discussion” section. For example, we propose that: 1) substrates may shift the equilibrium between oligomeric and monomeric species of DAT at the cell surface; 2) coat-free endocytosis may serve as part of the regulatory and quality-control mechanisms that remove excess DAT from the cell surface in the absence of substrates or clear DAT oligomers as they are functionally impaired (reduced functionality of DAT dimers/oligomers was proposed by Zhen and Reith, 2018); 3) cluster-triggered endocytosis may be common for other multimeric transmembrane proteins. We also emphasize that defining the molecular mechanisms of DAT interaction with AIM-100 and mapping the novel interaction interface in the DAT molecule may open avenues for developing new approaches for modulating DAT activities and subcellular localization at the highest specificity.

Please see the fourth and last paragraphs of the “Discussion”, and other parts of “Discussion”.

2) Along the same lines, because the effect of AIM-100 seems to be dependent on the inward facing conformation and the experiments are not conclusive about whether the compound binds to an allosteric site, it would be interesting to assess the effect of an inhibitor that preferentially binds an inward-facing conformation, such as ibogaine.

Unfortunately, ibogaine is a Schedule I controlled substance in US. While I am licensed to use Schedule II-IV compounds, obtaining the Schedule I license is a very tedious process and may take several months (I have applied two months ago). Therefore, instead of ibogaine, we tested the effects of modafinil, a compound that is available to us and that was proposed to bind DAT substrate binding sites and stabilize it in an occluded-intermediate and/or inward-facing conformation (Schmitt and Reith, 2011), similarly to what was proposed for the effects of ibogaine on DAT. We found that modafinil inhibits AIM-100-induced oligomerization and endocytosis of DAT. Please see the second paragraph of the subsection “DAT inhibitors and substrates diminish the effects of AIM-100 on DAT”, and a new Figure 6.

3) Also, does AIM-100 inhibit binding of a radiolabeled inhibitor to the transporter and, if so, is the inhibition competitive or allosteric in nature? This would further address the issue of how AIM-100 exerts its action on DAT.

Dr Melikian and co-workers previously demonstrated that AIM-100 inhibits CFT binding to DAT at 4^o^C with Ki 20-40 μM. We performed several binding saturation assays using the radiolabeled cocaine analog WIN34545 (CFT) in the absence or presence of AIM-100 at 20^o^C, and found that the inhibition is of non-competitive nature. The rationale for conducting experiments at 20^o^C but not in 4^o^C is based on our observation that AIM-100 efficiently increases DAT oligomerization at 20^o^C but not at 4^o^C. Because AIM-100 is highly lipophilic, it is poorly water-soluble in concentrations higher than 20 μM, which precluded testing the effects of high AIM-100 concentrations on CFT binding. Please see the last paragraph of the subsection “DAT inhibitors and substrates diminish the effects of AIM-100 on DAT” and new Figure 5—figure supplement 1.

Altogether the data with various substrates and inhibitors of DAT (Figures 5-6 and Figure 5—figure supplement 1) led to a new working model of how AIM-100 increases DAT oligomerization. This model is discussed as follows:

“The inhibitory action of DAT blockers and substrates on AIM-100-induced DAT oligomerization (Figures 5-6) suggest that AIM-100 effect may require binding of AIM-100 to a site in the DAT molecule that is inaccessible when the transporter is occupied by these substrates or inhibitors. […] The inability of AIM-100 to increase DAT oligomerization at 4^o^C (Figure 5—figure supplement 1) supports the hypothesis that an effective diffusion of AIM-100 across and/or within the membrane is involved in the AIM-100 action on DAT.”

4) Figure 9 should be supplemented. Do Ack knock-down/inhibition or knock-down/overexpression of K44A dominant negative affect the AIM-100 induced multimerization of DAT?

The lack of the effect of a potent Ack1 inhibitor KRCA-008 on DAT oligomerization is shown in new Figure 10—figure supplement 2. Because transient transfection of dynamin mutant results in its expression only in a small population of cells, biochemical experiments testing DAT oligomerization are not feasible. Therefore, we presented the blot demonstrating the lack of the effect of another robust inhibitor of clathrin-mediated endocytosis, clathrin heavy chain siRNA, on AIM-100-induced DAT oligomerization. Please see Figure 10—figure supplement 2.

5) Sorkin and colleagues were among the pioneers to use FRET microscopy in neurotransmitter transporters and revealed for the first time that DAT forms oligomers in living cells throughout the secretory pathway and at the cell surface. Given that these authors have such a strong background in fluorescence microscopy in the living cell, it is rather surprising that this methodology is not being used in the current manuscript where oligomerization is in the main focus. Hence, it is not clear to the reviewer why only biochemical assays were used to make one of the two main statements in this paper. At the very least, it would be desirable to see that AIM-100 is enhancing the oligomeric properties of DAT also under more physiological conditions, i.e. in living cells.

We thank reviewers for this idea. We were able to set up FRET measurements using spinning disk confocal system and test the effect of AIM-100 on CFP-DAT/YFP-DAT FRET in living cells with high resolution, so that FRET could be measured in individual structures, compartments and organelles. We observed a very clear AIM-100-induced increase in FRET intensities in the plasma membrane clusters and endosomes. Please see the last paragraph of the subsection “AIM-100 induced rapid nanoclustering of DAT on the cell surface” and Figure 3—figure supplement 1. These data together with the observation of rapid DAT clustering using high-resolution TIR-FM (Figure 3) support our conclusion that AIM-100 enhances DAT oligomerization in living cells.

6) The authors argue that an important control was unaffected by AIM-100: "Furthermore, AIM-100 did not inhibit recycling of the transferrin receptor (data not shown)." Here, my question is whether AIM-100 induces internalization of the TFR at all (which would also be a nice and independent control) and furthermore, to show these data.

The full-size images of transferrin conjugated to TexasRed (Tfn-TxR) in vehicle- and AIM-100-treated cells, which correspond to small insets in Figure 10, are now presented in new Figure 10—figure supplement 1. Quantification of the amount of Tfn-TxR in endosomes are presented in the same figure. Constitutive clathrin-mediated internalization of transferrin is very fast (20-25%/min), and certainly, AIM-100 does not accelerate this process. On the contrary, we noticed that AIM-100 partially inhibits the accumulation of Tfn-TxR in endosomes, possibly due to a potential impact of AIM-100 on the activities of Ack1 (and possibly other tyrosine kinases) that are proposed to be involved in the endosomal sorting. Please see the first paragraph of the subsection “Mechanisms of AIM-100-induced DAT trafficking”.